# The Influence of DOC on the Migration Forms of Elements and Their Sedimentation from River Waters at an Exploited Diamond Deposit (NW Russia)

Alexander I. Malov [1,*], Evgeniya S. Sidkina [2] and Elena V. Cherkasova [3]

[1] N. Laverov Federal Center for Integrated Arctic Research of the Ural Branch of the Russian Academy of Sciences, 20 Nikolsky Ave., 163020 Arkhangelsk, Russia

[2] Geological Institute of the Russian Academy of Sciences, 7 bld. 1. Pyzhevsky Lane, 119017 Moscow, Russia; sidkinaes@yandex.ru

[3] Vernadsky Institute of Geochemistry and Analytical Chemistry of the Russian Academy of Sciences, 19 Kosygin Str, 119991 Moscow, Russia; wri-lab@geokhi.ru

* Correspondence: malovai@yandex.ru

**Abstract:** The development of mineral deposits causes changes that are comparable to natural exogenous geological processes, and prevail over the latter in local areas of intensive mining activity. In this article, a diamond deposit is selected, developed by quarries of great depth, and a forecast is made of the impact of drainage water discharge on changes in the composition of surface water and bottom sediments during the entire period of development of the deposit. Modeling was performed according to various scenarios, taking into account changes in the total dissolved solids of groundwater from 0.5 to 21.7 g/kg $H_2O$. Thermodynamic calculations were carried out using the HCh software package. The role of dissolved organic carbon in the migration of chemical elements and the effect of DOC on the precipitation of chemical elements from mixed solutions is given. It has been established that fulvic acid completely binds to Fe in the $Fe(OH)_2FA^-$ complex in all types of natural waters and under all mixing scenarios. With humic acid, such a sharp competitive complex formation does not occur. It is distributed among the various elements more evenly. It was determined that the mass of precipitating iron in the presence of DOC decreases by 18–27%, and its precipitation in winter is more intense. In contrast to Fe, the precipitation of Ca, Mg, and C from solutions with DOC is higher in summer, and there are more of them in the solutions in winter. This study contributes to a better understanding of the behavior of heavy metals in surface waters and sediments under anthropogenic pressures in order to improve the sustainable management of water resources in the face of anthropogenic activities.

**Keywords:** wastewater; river water; groundwater; migration species; bottom sediments; modeling





## 1. Introduction

The development of mineral deposits causes changes that are comparable to natural exogenous geological processes, and prevail over the latter in local areas of intensive mining activity. The resulting technogenesis initiates an aggravation of the course of destructive geological processes and introduces substances, forces, and processes into the biosphere that change and disrupt its equilibrium functioning and the closed nature of the biotic cycle [1,2]. The strongest impact on the natural landscape is exerted by the open-pit mining of minerals, the area of which is constantly growing. Mining developments lead to the emergence of geochemical anomalies of anthropogenic origin when elevated concentrations of individual elements and their compounds are observed in mining areas [3–5]. They are especially pronounced in surface waters and bottom sediments of streams and reservoirs, into which wastewater from mining enterprises that extract coal and iron ores is discharged [6–12].

Mining operations at diamond deposits are no exception. However, due to the much smaller distribution of such deposits in the world and limited and (or) closed information,

there are relatively few scientific publications on their impact on the ecological state of water resources and their exploitation is more often analyzed in a socio-economic context [13–17]. At the same time, they are characterized by specific features associated, among other things, with the mineralogical composition of ultramafic kimberlite rocks, as shown by our studies at a diamond deposit in northwestern Russia [18,19] (Figure 1).

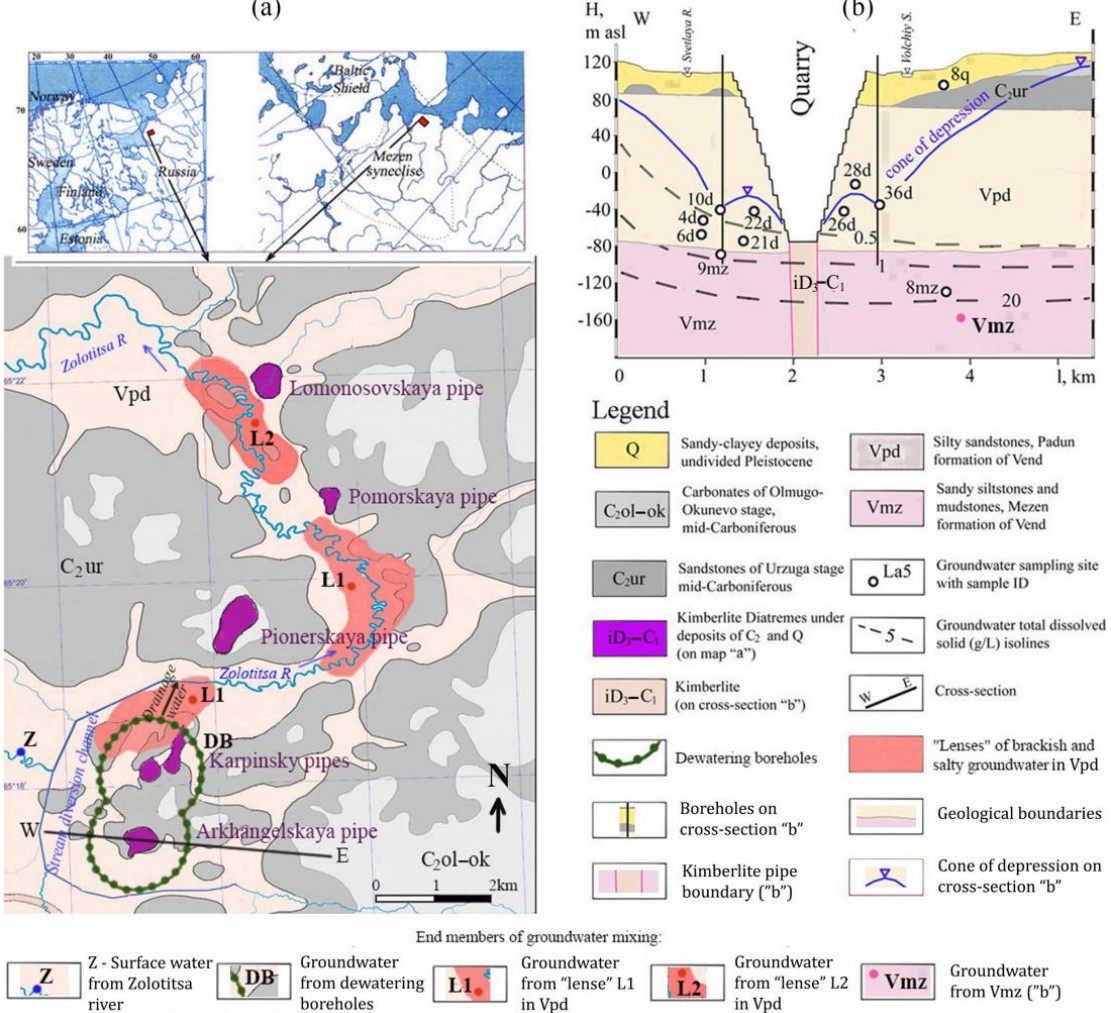

**Figure 1.** (**a**) General location of the study site showing the end members involved in the mixing of the surface waters of Zolotitsa River (Z) with fresh groundwater pumped from dewatering boreholes (DBs); brackish and salty groundwater pumped from "lenses" in the Vendian Padun Formation (L1 and L2); and (**b**) salty groundwater pumped from the Vendian Mezen Formation (Vmz).

The kimberlites of this deposit are characterized by very strong alteration associated with chemical weathering of the main rock-forming mineral olivine. This weathering took place both at the stage of the intrusion of high-temperature kimberlite melt into the flooded sedimentary cover, and during further water–rock interaction over 460 million years [20]. As a result, the overwhelming volume of kimberlite pipes is represented by a clay mineral from the montmorillonite group, saponite. When diamonds are extracted at processing plants, saponite is completely washed out with water, and in the form of a colloidal suspension is removed to the filtration fields in the swamp massif, on which it partially settles, and partially penetrates into the river and settles there. Possessing high sorption properties, it sorbs heavy metals entering the river with drainage water discharged into it from dewatering boreholes (DBs) (Figure 1a).

In addition, in the course of geological exploration, it was found that in the valley of the Zolotitsa River, saline water is being discharged from deep aquifers. Currently, they are

located in the form of saline lenses L1 and L2 in the aquifer of the Vendian Padun Formation (Vpd) sandstones (Figure 1a) and have already begun to pull up to the dewatering boreholes from lens L1 due to the cone of depression formed around the DBs (Figure 1b). In the future, the open pit field will expand to the north, and the process of drainage water salinization will be intensified due to lens L2, as well as due to the upwelling of saline water from the deep horizon of sandstones of the Vendian Mezen Formation (Vmz) (Figure 1b).

Salinization and (or) pollution of the Zolotitsa River are unacceptable due to its special fish protection status [21]; therefore, a detailed analysis of the formation of the chemical composition of surface water and bottom sediments in the process of mining operations at the deposit is necessary until they are fully completed. The water of the Zolotitsa River, like most of the rivers of the Northern region, contains a high concentration of dissolved organic carbon (DOC), which plays a significant role in the behavior of some chemical elements, including potentially toxic ones. When considering the issues of mixing natural waters of various types, it is necessary to take into account the presence of organic matter in the system for a correct understanding of the forms of occurrence of chemical elements and the possibility of secondary mineral formation. Thus, in the formation of complex particles of metals with organic matter, it is possible to block the formation of secondary mineral phases. In other words, the presence of DOC in water changes the functioning of geochemical barriers [22–24].

We have already considered this problem in previous studies [25] using the calculation of the equilibrium composition of mixed solutions. Our task was to estimate the maximum possible scale of sedimentation of secondary minerals from discharged waters. However, we did not take into account organic matter in the model, which led to insufficiently good comparability with natural observations in terms of elements that have a strong affinity for organic matter. Therefore, in this work, we carried out thermodynamic calculations taking into account DOC. The calculations were carried out using the HCh software package [26]. The main tasks were the following: (i) determination of the migration forms of chemical elements both in the initial natural surface and groundwater, and under various scenarios of changes in the composition of surface waters with the further development of mining operations; (ii) assessment of the role of DOC in the intensity of precipitation of chemical elements from mixed solutions of drainage and surface waters.

The following scenarios of river surface water mixing were considered: (i) with freshwater DBs, (ii) taking into account the inflow of saline water to the DBs from lens L1, (iii) taking into account the inflow of saline water to the DBs from lens L2, (iv) taking into account the inflow of saline water to the DBs from the aquifer of the Vendian Mezen Formation (Vmz).

## 2. Materials and Methods

### 2.1. Natural Conditions of the Study Area

The diamond deposit is located in the northwest of Russia, in the area located at the junction of the Baltic Shield and the Mezen Syneclise (Figure 1a).

The study area is represented by six kimberlite pipes: Arkhangelskaya, Karpinsky-1, Karpinsky-2, Pionerskaya, Pomorskaya, and Lomonosovskaya (see Figure 1a). The thickness of the sedimentary cover is about 1 km. The study site is composed of sequences of different ages, namely, the Middle Carboniferous carbonate-terrigenous ($C_2$), Vendian (Ediacaran), Padun (Vpd), Mezen (Vmz), and Ust-Pinega (Vup) formations. The Vendian terrigenous sequence is composed of alternating sandstones, siltstones, and mudstones. The carbonate-terrigenous rocks are overlain by a 10 to 15 m thick sequence of Valdaian (Weichselian) glaciation sediments (Q), which are largely represented by moraine boulder loams with localities of fluvioglacial and glaciolacustrine sands.

The depths of the quarries to date are 200 m on the Arkhangelskaya pipe (Figure 1b) and 170 m on the Karpinsky-1 pipe. Dewatering of the quarry by surface pumps directly from the pit was started in 2003, and preliminary drainage of rock mass by a system of drainage wells was started in 2005. The protection of quarries from groundwater is

carried out by an external drainage circuit of 70 drainage boreholes with a total flow rate of 5000 m$^3$/h. At the same time, the productivity of the quarry dewatering by surface pumps on the Arkhangelskaya pipe is 1000 m$^3$/h, and on the Karpinsky-1 pipe is 300 m$^3$/h. The Zolotitsa River flowing through the area of the deposit is diverted to the west by means of an artificial canal. Drainage water from dewatering wells is discharged into this channel. The quarry waters are discharged and purified on the filtration fields before being dumped into the river.

The hydrograph of the river is shown in Figure 2.

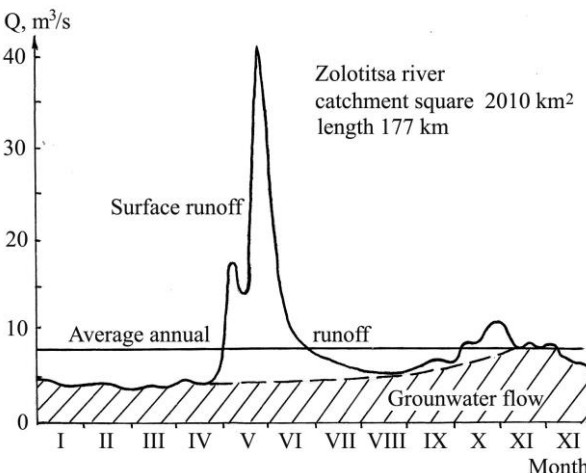

**Figure 2.** Hydrograph at the mouth of the Zolotitsa River.

The main role in the flooding of the deposit is played by the aquifer complex of the host rocks of the Padun stage of the Vendian; the thickness of the complex within the study area is 160–180 m, and hydraulic conductivity or coefficient of permeability (k$_f$) is 1.5–1.7 m/day. The complex is unconfined laterally; it is represented by an uneven alternation of layers of sandstones, siltstones, and, to a lesser extent, mudstones. A subordinate role is played by the upper overlapping deposits, which combine the aquifers of the Quaternary sediments and the Urzuga stage of the Middle Carboniferous, composed of weakly-cemented fine-grained clay sandstones with a thickness of 25–30 m (k$_f$ 1.0 m/day). At the base of the cross-section, there is a sequence of water-bearing complexes of Mezen (k$_f$ 0.03 m/day) and Ust-Pinega (k$_f$ $10^{-3}$–$10^{-4}$ m/day) formations, which are characterized by an uneven alternation of siltstones and argillites with rare thin sandstone interlayers. In relation to the Padun complex, this is an aquitard. The explosion pipes break through the Vendian deposits and overlap with rocks of the Carboniferous and Quaternary ages. In them, the upper crater part is formed with a thickness of 110–150 m, composed of tuff and tuffites, and the lower crater part is represented by kimberlite breccia. For the upper crater facies, an increased permeability (k$_f$~0.6–0.9 m/day) is typical, but for the lower crater facies, permeability is much lower (k$_f$~0.02 m/day).

As noted above, in the aquifer of the Vendian Padun Formation (Vpd) among fresh waters, a discharge of saline waters with TDS of 2–10 g/L was established from deep aquifers. Currently, they are located in the form of saline "lenses" L1 and L2 (Figure 1a). Even more saline waters with TDS over 20 g/L are located deeper in the aquifer of the sandstones of the Vendian Mezen Formation (Vmz) (Figure 1b).

*2.2. Methods*

Modeling of changes in the composition of water in the Zolotitsa River (Z) was carried out when fresh drainage groundwater was discharged into it from a DB system according to the following scenarios: (1) mixing of DB–Z in a ratio of 2:1; (2) the same, when brackish waters are drawn to the DBs from the lens in the aquifer complex of the deposits of the Vendian Padunskaya suite (L1) in a ratio of 1:3; (3) the same, when saline water is pulled up to the DBs from the lens in the aquifer complex of the deposits of the Vendian Padunskaya

suite (L2) in a ratio of 1:3; (4–6) the same, when saline waters are drawn to the DBs from the underlying aquifer of the Vendian Mezen Formation (Vmz) in ratios of 1:100, 4:100, and 7:100. The ratio 7:100 is the ratio of the water conductivities of the aquifers of the Mezen and Padun suites. Theoretically, in this ratio, the maximum flow in the steady state can be carried out.

Thermodynamic calculations were carried out using the HCh software package [26]. Analytically determined concentrations of chemical elements were given as input data to the model. The equilibrium compositions of natural waters and mixed solutions were calculated in the proportions described above. The equilibria in the program are calculated by the Gibbs free-energy minimization algorithm. The calculation of the activity coefficients of the components of the aqueous solution is carried out according to the Debye–Huckel model. The system consisted of 24 chemical elements: O, H, C, S, Cl, Ca, Mg, Na, K, Fe, Al, Mn, Ni, Cu, Zn, Cd, Cr, Sr, Pb, Mo, As, U, FA (fulvic acid), and HA (humic acid). FA and HA are introduced into the HCh program as new independent elements. To take into account the complex formation of chemical elements with organic ligands, the Unitherm database was supplemented with the corresponding free energies. The free energies $FA^{2-}$ and $HA^-$ are taken equal to zero. The free energies of the formation of complexes with organic ligands are calculated by the equation:

$$G^{0*}_{fMeA} = G^0_{fMe^{m+}} - RTlnK_{ef}$$

where $G^{0*}_{fMeA}$ is the free energy of the complex formation with an organic anion; $G^0_{fMe^{m+}}$ is the free energy of the metal ion; $R$ is the gas constant; $T$ is the temperature (298.15 K); $K_{ef}$ is the effective stability constant [27]. The effective stability constants from [28–38] were used to calculate the free energies of the formation of complexes with fulvic and humic acids. A complete list of aqueous species is given in Table 1.

Accounting for organic matter in computer simulations has been successfully performed in other objects [39–41].

The model took into account the possibility of the formation of the following minerals: dolomite, gibbsite, goethite, pyrolusite, ankerite, malachite, siderite, and zincite. In addition to the above phases, the model took into account the possibility of the formation of calcite containing strontium, zinc, lead, and manganese. In other words, it is a solid solution of variable composition.

The chemical compositions of the water samples were taken as the initial data (Table 2). The procedures by which they were obtained are described in [18]. All of the water samples were filtered through 0.45 μm acetate cellulose in the field. The solutions that were filtered for cation and trace element analyses were acidified with double-distilled $HNO_3$ (pH < 2); the samples for anion analysis were not acidified. The water temperature, pH, and Eh were measured in the field using portable HANNA instruments with uncertainties of 0.1 °C, 0.05 pH units, and 0.1 mV, respectively. The calcium, magnesium, sodium, and potassium concentrations were determined with an uncertainty of 1–2% by using an atomic absorption spectrometer (AAS) (Perkin-Elmer 5100 PC, USA). The alkalinity was measured by potentiometric titration with HCl using an automated titrator (Metrohm 716 DMS Titrino, Switzerland) and using the Gran method (detection limit $10^{-5}$ M; uncertainty at ≥0.5 mmol/L 1–3%, and at <0.5 mmol/L 7%). The major anion concentrations ($Cl^-$, $SO_4^{2-}$) were measured by ion chromatography (HPLC, Dionex ICS 2000, Switzerland) with an uncertainty of 2%. Major and trace elements were determined without pre-concentration by inductively coupled plasma mass spectrometry (ICP-MS) (Agilent 7500ce) at GET, Toulouse, France. Good agreement (≤10%) between the measured and certified concentrations in a certified river water sample (SLRS-5) was achieved. DOC was measured using a Shimadzu TOC 6000 (Japan) with an uncertainty of 5%. The contents of fulvic and humic acids were calculated from the DOC concentration according to the method proposed in [27]. According to the method, it is assumed that carbon is 40% of fulvic and humic acids, and their ratio is 10:1.

**Table 1.** Aqueous species taken into account in the simulation.

| $H_2O$ | $CdCl_3^-$ | $Fe^{2+}$ | $MnO^0$ | $PbCO_3^0$ | $UO_2^{2+}$ | $Cu(OH)_2FA^{2-}$ |
|---|---|---|---|---|---|---|
| $H^+$ | $CdCl_4^{2-}$ | $FeOH^+$ | $HMnO_2^-$ | $PbHCO_3^+$ | $UO_2OH^+$ | $ZnFA^0$ |
| $OH^-$ | $Cd(HSO_4)_2^0$ | $FeO^0$ | $MnO_2^{2-}$ | $SO_4^{2-}$ | $U_2O_4(OH)_2^{2-}$ | $PbFA^0$ |
| $H_2$ (aq) | $CdHCO_3^+$ | $HFeO_2^-$ | $MnCl^+$ | $HSO_4^-$ | $UO_2(OH)_2^0$ | $HA^-$ |
| $O_2$ (aq) | $Cl^-$ | $FeCl^+$ | $MnCl_2^0$ | $Sr^{2+}$ | $UO_2SO_4^0$ | $HHA^0$ |
| $Al^{3+}$ | $HCl^0$ | $FeCl_2^0$ | $MnSO_4^0$ | $SrOH^+$ | $UO_2(SO_4)_2^{2-}$ | $CaHA^+$ |
| $AlOH^{2+}$ | $Cr^{2+}$ | $FeSO_4^0$ | $MnHCO_3^+$ | $SrCl^+$ | $UO_2CO_3^0$ | $MgHA^+$ |
| $AlO^+$ | $Cr^{3+}$ | $FeCO_3^0$ | $Mn^{3+}$ | $SrCl_2^0$ | $UO_2(CO_3)_2^{2-}$ | $FeHA_3^0$ |
| $AlOOH^0$ | $CrO^+$ | $Fe^{3+}$ | $MnO_4^-$ | $SrSO_4^0$ | $UO_3^0$ | $CuHA_2^0$ |
| $AlO_2^-$ | $CrOH^{2+}$ | $FeOH^{2+}$ | $MnO_4^{2-}$ | $SrCO_3^0$ | $UO_4^{2-}$ | $CuHA^+$ |
| $HAsO_2^0$ | $HCrO_2^0$ | $FeO^+$ | $MoO_4^{2-}$ | $SrHCO_3^+$ | $HUO_4^-$ | $ZnHA^+$ |
| $AsO_2^-$ | $CrO_2^-$ | $HFeO_2^0$ | $HMoO_4^-$ | $U^{3+}$ | $Zn^{2+}$ | $PbHA_2^0$ |
| $HAsO_3^{2-}$ | $CrO_4^{2-}$ | $FeO_2^-$ | $Na^+$ | $UOH^{2+}$ | $ZnOH^+$ | $PbHA^+$ |
| $AsO_4^{3-}$ | $HCrO_4^-$ | $FeCl^{2+}$ | $NaOH^0$ | $UO^+$ | $ZnO^0$ | $H_2FA^0$ |
| $HAsO_4^{2-}$ | $Cr_2O_7^{2-}$ | $FeCl_2^+$ | $NaCl^0$ | $HUO_2^0$ | $HZnO_2^-$ | $MnFA^0$ |
| $H_2AsO_4^-$ | $Cu^+$ | $FeCl_3^0$ | $NaSO_4^-$ | $U^{4+}$ | $ZnO_2^{2-}$ | $CdFA^0$ |
| $H_3AsO_4^0$ | $CuOH^0$ | $FeSO_4^+$ | $NaCO_3^-$ | $UOH^{3+}$ | $ZnCl^+$ | $MnHA^+$ |
| $CO_3^{2-}$ | $CuCl^0$ | $FeHSO_4^{2+}$ | $NaHCO_3^0$ | $UO^{2+}$ | $ZnCl_2^0$ | $CdHA_2^0$ |
| $HCO_3^-$ | $CuCl_2^-$ | $K^+$ | $Ni^{2+}$ | $HUO_2^+$ | $ZnCl_3^-$ | $MoO_2HA^+$ |
| $CO_2$ (aq) | $CuCl_3^{2-}$ | $KOH^0$ | $NiOH^+$ | $UO_2^0$ | $ZnCl_4^{2-}$ | $MoO_2HA_2^0$ |
| $Ca^{2+}$ | $CuHCO_3^0$ | $KCl^0$ | $NiO^0$ | $HUO_3^-$ | $ZnSO_4^0$ | $UO_2FA^0$ |
| $CaOH^+$ | $Cu^{2+}$ | $KSO_4^-$ | $HNiO_2^-$ | $UCl^{3+}$ | $ZnHSO_4^+$ | $UO_2FA_2^{2-}$ |
| $CaCl^+$ | $CuOH^+$ | $KHSO_4^0$ | $NiO_2^{2-}$ | $UCl_2^{2+}$ | $ZnCO_3^0$ | $UO_2HA^+$ |
| $CaCl_2^0$ | $CuO^0$ | $KCO_3^-$ | $NiCl^+$ | $USO_4^{2+}$ | $ZnHCO_3^+$ | $UO_2HA_2^0$ |
| $CaSO_4^0$ | $HCuO_2^-$ | $KHCO_3^0$ | $Pb^{2+}$ | $UHSO_4^{3+}$ | $FA^{2-}$ | $SrFA^0$ |
| $CaCO_3^0$ | $CuO_2^{2-}$ | $Mg^{2+}$ | $PbOH^+$ | $UCO_3^{2+}$ | $HFA^-$ | $SrHu^+$ |
| $CaHCO_3^+$ | $CuCl^+$ | $MgOH^+$ | $PbO^0$ | $UHCO_3^{3+}$ | $CaFA^0$ | $Sr(HA)_2^0$ |
| $Cd^{2+}$ | $CuCl_2^0$ | $MgCl^+$ | $HPbO_2^-$ | $UO_2^+$ | $MgFA^0$ | $CrFA^+$ |
| $CdOH^+$ | $CuCl_3^-$ | $MgCl_2^0$ | $PbCl^+$ | $UO_2OH^0$ | $FeFA^0$ | $CrHu^{+2}$ |
| $CdO^0$ | $CuCl_4^{2-}$ | $MgSO_4^0$ | $PbCl_2^0$ | $UO_3^-$ | $FeFA^+$ | $Cr(OH)FA^0$ |
| $HCdO_2^-$ | $CuSO_4^0$ | $MgCO_3^0$ | $PbCl_3^-$ | $UO_2Cl^0$ | $FeOHFA^0$ | $NiFA^0$ |
| $CdO_2^{2-}$ | $CuHSO_4^+$ | $MgHCO_3^+$ | $PbCl_4^{2-}$ | $UO_2Cl_2^-$ | $Fe(OH)_2FA^-$ | $NiHA^+$ |
| $CdCl^+$ | $CuCO_3^0$ | $Mn^{2+}$ | $PbSO_4^0$ | $UO_2HCO_3^0$ | $AlFA^+$ | |
| $CdCl_2^0$ | $CuHCO_3^+$ | $MnOH^+$ | $PbHSO_4^+$ | $UO_2(HCO_3)_2^-$ | $CuFA^0$ | |

**Table 2.** Initial compositions of surface water and groundwater, taken in modeling their mixing.

| | ZS [1] | ZW [1] | DBs | L1 | L2 | Vmz |
|---|---|---|---|---|---|---|
| T °C | 14 | 0.6 | 4.5 | 4.9 | 5.7 | 6.9 |
| pH | 7.5 | 7.3 | 8.6 | 7.9 | 7.7 | 8.3 |
| Eh, mV | 281 | 120 | 65 | −22 | −38 | −80 |
| mg/kg $H_2O$ | | | | | | |
| Fulvic acids (FA) | 28.2 | 16.1 | 3.03 | 7.55 | 2.92 | 2.56 |
| Humic acids (HA) | 1.57 | 0.9 | 0.17 | 0.42 | 0.16 | 0.14 |
| $O_2$ | 10.3 | 6.1 | 1.7 | 0 | 1.2 | 0 |
| Na | 13.3 | 13.3 | 101.3 | 792 | 1960 | 5374 |
| Mg | 3.12 | 3.12 | 9.86 | 48.4 | 298 | 484 |
| K | 0.8 | 0.8 | 4.24 | 6.88 | 33.6 | 52.8 |
| Ca | 6.06 | 6.06 | 17.6 | 49.6 | 495 | 1804 |
| Cl | 8.22 | 8.22 | 73 | 1009 | 3034 | 11,502 |
| $HCO_3^-$ | 48.8 | 48.8 | 211 | 325 | 255 | 19.8 |
| $SO_4^{2-}$ | 4.2 | 4.2 | 33.6 | 292 | 2323 | 2326 |
| TDS | 87.1 | 87.1 | 455 | 2528 | 8418 | 21,664 |
| µg/kg $H_2O$ | | | | | | |
| Al | 57.2 | 57.2 | 7 | 25.2 | 34 | 187 |

**Table 2.** *Cont.*

|  | ZS [1] | ZW [1] | DBs | L1 | L2 | Vmz |
|---|---|---|---|---|---|---|
| Cr | 0.37 | 0.37 | 0.63 | 0.99 | 1.54 | 18.5 |
| Mn | 13.4 | 13.4 | 12.5 | 78.6 | 814 | 3153 |
| Fe | 339 | 339 | 35.1 | 1341 | 1872 | 6564 |
| Ni | 0.3 | 0.3 | 0.11 | 0.82 | 1.86 | 2.31 |
| Cu | 0.25 | 0.25 | 0.16 | 1.2 | 1.84 | 0.29 |
| Zn | 5.13 | 5.13 | 2.17 | 14.8 | 23.8 | 49.2 |
| As | 0.5 | 0.5 | 0.6 | 0.39 | 0.36 | 1.22 |
| Sr | 64.3 | 64.3 | 168 | 671 | 12,301 | 38,594 |
| Mo | 0.37 | 0.37 | 2.7 | 18.1 | 3.46 | 9.3 |
| Cd | 0.0028 | 0.0028 | 0.0044 | 0.04 | 0.04 | 0.035 |
| Pb | 0.106 | 0.106 | 0.045 | 1.47 | 2.02 | 0.06 |
| U | 0.59 | 0.59 | 6.78 | 1.57 | 15.2 | 0.15 |

Note: [1] S—summer; W—Winter.

## 3. Results and Discussion

### 3.1. Organic and Inorganic Aqueous Species of Elements in the Natural Waters of the Study Area

The results of calculations of the chemical element speciation in the natural waters of the study area are given in Table 3.

**Table 3.** Simulated dominant aqueous species of chemical elements in natural waters at the study area (mol %).

| Aqueous Species | ZS | ZW | DBs | L1 | L2 | Vmz | Aqueous Species | ZS | ZW | DBs | L1 | L2 | Vmz |
|---|---|---|---|---|---|---|---|---|---|---|---|---|---|---|
| **FA** | | | | | | | $CuCl_2^-$ | 0 | 0 | 0 | 80.86 | 57.38 | 24.38 |
| $Fe(OH)_2 FA^-$ | 100 | 100 | 100 | 100 | 100 | 100 | $CuCl_3^{2-}$ | 0 | 0 | 0 | 17.60 | 42.26 | 75.58 |
| **HA** | | | | | | | $Cu^{2+}$ | 4.77 | 10.17 | 0.77 | 0 | 0 | 0 |
| $HA^-$ | 40.67 | 40.67 | 34.49 | 12.91 | 2.27 | 0.86 | $CuOH^+$ | 1.99 | 2.82 | 0.55 | 0 | 0 | 0 |
| $HHA^0$ | 0.01 | 0.01 | 0 | 0.01 | 0 | 0 | $CuO^0$ | 12.86 | 11.97 | 7.34 | 0 | 0 | 0 |
| $MgHA^+$ | 15.02 | 15.02 | 14.72 | 32.27 | 25.33 | 14.49 | $CuCl^+$ | 0 | 0.01 | 0 | 0 | 0 | 0 |
| $CaHA^+$ | 43.94 | 43.94 | 50.74 | 54.64 | 72.29 | 84.37 | $CuSO_4^0$ | 0 | 0.06 | 0.02 | 0 | 0 | 0 |
| $MnHA^+$ | 0 | 0 | 0 | 0.12 | 0.08 | 0.22 | $CuCO_3^0$ | 80.19 | 74.63 | 91.31 | 0 | 0 | 0 |
| $CuHA^+$ | 0.01 | 0.01 | 0 | 0 | 0 | 0.02 | $CuHCO_3^+$ | 0.02 | 0.03 | 0.01 | 0 | 0 | 0 |
| $ZnHA^+$ | 0.33 | 0.33 | 0.05 | 0.04 | 0.02 | 0 | $Cu(OH)_2 FA^{2-}$ | 0 | 0 | 0 | 0 | 0 | 0 |
| $SrHA^+$ | 0.01 | 0.01 | 0 | 0.01 | 0.01 | 0.04 | $CuHA^+$ | 0.14 | 0.31 | 0 | 0 | 0 | 0 |
| $PbHA^+$ | 0 | 0 | 0 | 0 | 0 | 0 | **Zn** | | | | | | |
| **Mg** | | | | | | | $Zn^{2+}$ | 70.70 | 80.12 | 44.46 | 76.64 | 73.22 | 83.25 |
| $Mg^{2+}$ | 98.49 | 98.82 | 93.92 | 86.50 | 71.95 | 78.88 | $ZnOH^+$ | 19.02 | 14.32 | 16.01 | 2.70 | 0.82 | 0.65 |
| $MgOH^+$ | 0.01 | 0 | 0.01 | 0 | 0 | 0 | $ZnO^0$ | 0.07 | 0.03 | 0.11 | 0 | 0 | 0 |
| $MgCl^+$ | 0.01 | 0.01 | 0.11 | 1.00 | 1.8 | 5.87 | $HZnO_2^-$ | 0 | 0 | 0 | 0 | 0 | 3.62 |
| $MgCl_2^0$ | 0 | 0 | 0 | 0.05 | 0.23 | 2.49 | $ZnCl^+$ | 0.01 | 0.01 | 0.03 | 0.52 | 1.07 | 0.85 |
| $MgSO_4^0$ | 0.55 | 0.59 | 2.89 | 9.32 | 25.05 | 12.7 | $ZnCl_2^0$ | 0 | 0 | 0 | 0.02 | 0.08 | 0.21 |
| $MgCO_3^0$ | 0.18 | 0.08 | 1.04 | 0.17 | 0.02 | 0 | $ZnCl_3^-$ | 0 | 0 | 0 | 0 | 0.01 | 0.05 |
| $MgHCO_3^+$ | 0.76 | 0.5 | 2.04 | 2.96 | 0.95 | 0.06 | $ZnHS^+$ | 0 | 0 | 0 | 0.02 | 0.01 | 0.01 |
| $MgHA^+$ | 0 | 0 | 0 | 0 | 0 | 0 | $ZnSO_4^0$ | 0.41 | 0.49 | 1.38 | 7.97 | 23.06 | 11.27 |
| **Ca** | | | | | | | $ZnCO_3^0$ | 9.33 | 4.61 | 37.52 | 10.86 | 1.29 | 0.06 |
| $Ca^{2+}$ | 98.49 | 98.93 | 94.05 | 89.96 | 81.24 | 88.71 | $ZnHCO_3^+$ | 0.30 | 0.23 | 0.48 | 1.26 | 0.44 | 0.03 |
| $CaCl^+$ | 0.01 | 0.01 | 0.06 | 0.57 | 1.05 | 3.19 | $ZnHA^+$ | 0.16 | 0.19 | 0.01 | 0.01 | 0 | 0 |
| $CaCl_2^0$ | 0 | 0 | 0 | 0.01 | 0.03 | 0.3 | **As** | | | | | | |
| $CaSO_4^0$ | 0.36 | 0.38 | 1.93 | 6.15 | 16.68 | 7.74 | $AsO_4^{3-}$ | 0.02 | 0.01 | 0.06 | 0 | 0 | 0 |
| $CaCO_3^0$ | 0.35 | 0.15 | 1.87 | 0.30 | 0.03 | 0 | $HAsO_4^{2-}$ | 94.87 | 92.22 | 98.16 | 0 | 0 | 0 |
| $CaHCO_3^+$ | 0.78 | 0.52 | 2.09 | 3.01 | 0.97 | 0.06 | $H_2AsO_4^-$ | 5.11 | 7.77 | 1.78 | 0 | 0 | 0 |
| $CaHA^+$ | 0.01 | 0.01 | 0 | 0 | 0 | 0 | $HAsO_2^0$ | 0 | 0 | 0 | 98.92 | 99.55 | 99.62 |

**Table 3.** *Cont.*

| Aqueous Species | ZS | ZW | DBs | L1 | L2 | Vmz |
|---|---|---|---|---|---|---|
| **Al** | | | | | | |
| $Al^{3+}$ | 0 | 0 | 0 | 0 | 0.02 | 0.07 |
| $AlOH^{2+}$ | 0 | 0 | 0 | 0.02 | 0.38 | 0.9 |
| $AlO^{+}$ | 0.03 | 0.07 | 0.01 | 0.38 | 2.13 | 3.14 |
| $AlOOH^{0}$ | 5.49 | 8.22 | 2.29 | 14.60 | 28.71 | 32.13 |
| $AlO_2^{-}$ | 94.48 | 91.71 | 97.7 | 85.00 | 68.76 | 63.76 |
| **Cr** | | | | | | |
| $Cr^{3+}$ | 0 | 0 | 0 | 0.02 | 0.15 | 0.26 |
| $CrO^{+}$ | 0 | 0 | 0 | 87.73 | 70.35 | 60.12 |
| $CrOH^{2+}$ | 0 | 0 | 0 | 10.59 | 29.02 | 39.3 |
| $HCrO_2^{0}$ | 0 | 0 | 0 | 1.63 | 0.47 | 0.32 |
| $CrO_2^{-}$ | 0 | 0 | 0 | 0.01 | 0 | 0 |
| $CrO_4^{2-}$ | 0 | 95.88 | 99.13 | 0 | 0 | 0 |
| $HCrO_4^{-}$ | 0 | 4.12 | 0.87 | 0 | 0 | 0 |
| $CrHA^{2+}$ | 0 | 0 | 0 | 0.02 | 0.01 | 0 |
| **Mn** | | | | | | |
| $Mn^{2+}$ | 96.83 | 98.49 | 83.8 | 92.23 | 87.51 | 91.92 |
| $MnOH^{+}$ | 0.06 | 0.04 | 0.08 | 0.01 | | |
| $MnCl^{+}$ | 0.01 | 0.01 | 0.06 | 0.59 | 1.16 | 3.48 |
| $MnSO_4^{0}$ | 0.20 | 0.22 | 0.95 | 3.51 | 10.06 | 4.53 |
| $MnHCO_3^{+}$ | 1.01 | 0.68 | 2.2 | 3.66 | 1.27 | 0.07 |
| $MnO_4^-$ | 1.87 | 0.54 | 12.9 | 0 | 0 | 0 |
| $MnO_4^{--}$ | | 0 | 0.01 | 0 | 0 | 0 |
| $MnHA^{+}$ | 0.02 | 0.02 | 0 | 0 | 0 | 0 |
| **Fe** | | | | | | |
| $Fe^{2+}$ | 0 | 0 | 0 | 14.11 | 62.3 | 86.98 |
| $FeOH^{+}$ | 0 | 0 | 0 | 0.03 | 0.05 | 0.04 |
| $FeCl^{+}$ | 0 | 0 | 0 | 0.14 | 1.23 | 4.75 |
| $FeSO_4^{0}$ | 0 | 0 | 0 | 0.95 | 12.66 | 7.63 |
| $FeCO_3^{0}$ | 0 | 0 | 0 | 1.92 | 1.05 | 0.06 |
| $Fe(OH)_2 FA^{-}$ | 100 | 100 | 100 | 82.85 | 22.71 | 0.54 |
| **Ni** | | | | | | |
| $Ni^{2+}$ | 99.96 | 99.96 | 99.92 | 99.86 | 99.74 | 99.27 |
| $NiOH^{+}$ | 0.04 | 0.04 | 0.07 | 0.01 | | |
| $NiCl^{+}$ | 0 | 0 | 0.01 | 0.13 | 0.26 | 0.73 |
| $NiHA^{+}$ | 0 | 0 | 0 | 0 | 0 | 0 |
| **Cu** | | | | | | |
| $Cu^{+}$ | 0 | 0 | 0 | 0.19 | 0.02 | 0 |
| $CuOH^{0}$ | 0 | 0 | 0 | 0.04 | 0 | 0 |
| $CuCl^{0}$ | 0 | 0 | 0 | 1.31 | 0.34 | 0.04 |

| Aqueous Species | ZS | ZW | DBs | L1 | L2 | Vmz |
|---|---|---|---|---|---|---|
| $AsO_2^{-}$ | 0 | 0 | 0 | 1.08 | 0.45 | 0.38 |
| **Sr** | | | | | | |
| $Sr^{2+}$ | 98.19 | 98.44 | 92.86 | 83.30 | 66.76 | 81.37 |
| $SrCl^{+}$ | 0.01 | 0.01 | 0.07 | 0.55 | 0.87 | 2.8 |
| $SrCl_2^{0}$ | 0 | 0 | 0 | 0 | 0.01 | 0.15 |
| $SrSO_4^{0}$ | 0.90 | 0.97 | 4.71 | 13.71 | 31.68 | 15.64 |
| $SrCO_3^{0}$ | 0.11 | 0.05 | 0.59 | 0.09 | 0.01 | 0 |
| $SrHCO_3^{+}$ | 0.79 | 0.53 | 1.77 | 2.35 | 0.67 | 0.04 |
| **Mo** | | | | | | |
| $MoO_4^{2-}$ | 99.98 | 99.97 | 99.99 | 99.96 | 99.92 | 99.92 |
| $HMoO_4^{-}$ | 0.02 | 0.03 | 0.01 | 0.04 | 0.08 | 0.08 |
| **Cd** | | | | | | |
| $Cd^{2+}$ | 97.36 | 97.49 | 83.85 | 35.39 | 19.95 | 5.7 |
| $CdOH^{+}$ | 0.24 | 0.16 | 0.32 | 0.01 | | |
| $CdCl^{+}$ | 2.10 | 2.15 | 15.02 | 57.58 | 62.44 | 46.01 |
| $CdCl_2^{0}$ | 0 | 0 | 0.15 | 6.42 | 16.15 | 36.38 |
| $CdCl_3^{-}$ | 0 | 0 | 0 | 0.19 | 1.33 | 10.39 |
| $CdCl_4^{2-}$ | 0 | 0 | 0 | 0 | 0.05 | 1.52 |
| $CdHCO_3^{+}$ | 0.30 | 0.2 | 0.66 | 0.41 | 0.08 | 0 |
| **Pb** | | | | | | |
| $Pb^{2+}$ | 1.03 | 1.74 | 0.25 | 1.87 | 9.59 | 16.25 |
| $PbOH^{+}$ | 64.95 | 73 | 40.07 | 26.53 | 36.86 | 34.8 |
| $PbO^{0}$ | 0.01 | 0.01 | 0.01 | | | |
| $PbCl^{+}$ | 0.01 | 0.01 | 0.01 | 0.64 | 6.27 | 27.01 |
| $PbCl_2^{0}$ | 0 | 0 | 0 | 0.04 | 0.99 | 13.39 |
| $PbCl_3^{-}$ | 0 | 0 | 0 | 0 | 0.04 | 1.88 |
| $PbCl_4^{2-}$ | 0 | 0 | 0 | 0 | 0 | 0.78 |
| $PbHS^{+}$ | 0 | 0 | 0 | 0.01 | 0.02 | 0.01 |
| $PbSO_4^{0}$ | 0.01 | 0.02 | 0.02 | 0.36 | 5.18 | 3.44 |
| $PbCO_3^{0}$ | 33.84 | 24.97 | 59.64 | 70.52 | 41.02 | 2.44 |
| $PbHCO_3^{+}$ | 0 | 0 | 0 | 0.01 | 0.03 | 0 |
| $PbHA_2^{0}$ | 0 | 0 | 0 | 0 | 0 | 0 |
| $PbHA^{+}$ | 0.15 | 0.25 | 0 | 0.02 | 0 | 0 |
| **U** | | | | | | |
| $UO_2OH^{+}$ | 0.02 | 0.06 | 0 | 99.99 | 99.96 | 100 |
| $UO_2CO_3^{0}$ | 1.84 | 5.11 | 11.52 | 0 | 0 | 0 |
| $UO_2(CO_3)_2^{2-}$ | 55.66 | 57.18 | 0 | 0 | 0 | 0 |
| $UO_2(CO_3)_3^{4-}$ | 26.98 | 12.16 | 88.19 | 0 | 0.01 | 0 |
| $UO_3^{0}$ | 14.32 | 24.48 | 0.20 | 0.01 | 0.03 | 0 |
| $HUO_4^{-}$ | 1.18 | 1.01 | 0.04 | 0 | 0 | 0 |

### 3.1.1. Fulvic Aqueous Species

Fulvic acids ($FA^{2-}$) are fully bound to Fe in the $Fe(OH)_2FA^{-}$ in all types of natural surface and groundwater ZS, ZW, DBs, L1, L2, and Vmz (Figure 3a).

Fe has a high affinity for organic matter and ranks first among competitive complex formations in all landscape-climatic zones [42]. Accordingly, when calculating, it is complexed with $FA^{2-}$ in the first place, taking from the balance of the FA entered into the model everything that can be pulled onto itself. The $FA^{2-}$ remaining in the solution is generally distributed over other metals, but in our case, this is not observed, due to the rather high content of Fe.

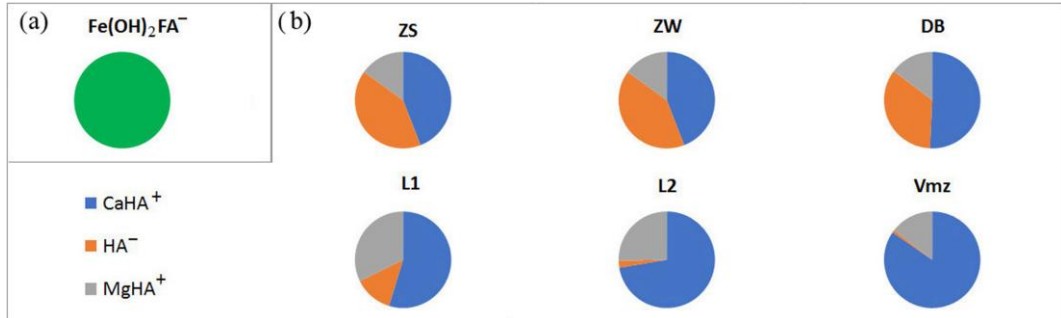

**Figure 3.** Main aqueous species of (**a**) fulvic acids and (**b**) humic acids in the natural waters of the study area (%).

The high content of iron in fresh oxygen waters with positive values of the redox potential is due to the oxidation of $Fe^{2+} \rightarrow Fe^{3+} + e$ and the subsequent formation of stable complex compounds of $Fe^{3+}$ with $FA^{2-}$. In our case, the hydroxofulvate complex $Fe(OH)_2FA^-$ turned out to be the most competitive. Organo-mineral species of Fe are very stable. The formation of these complexes does not allow the precipitation of Fe in the form of hydroxide [43]. According to calculations, after the FA specified in the balance is exhausted, Fe precipitates in the form of goethite. That is, for correct calculations of the amount of precipitated Fe minerals, it is critical to correctly determine the amount of organic matter in natural waters. In fresh waters, the ZS, ZW, and DB iron concentrations adjusted for deposition (see Tables 2 and 4) are 315, 180, and 33.84 μg/kg $H_2O$, respectively, and are limited by $FA^{2-}$ concentrations of 28.2, 16.1, and 3.03 mg/kg $H_2O$ (see Table 2), based on the ratio FA/Fe = 5000/55.84. In brackish and saline waters L1, L2, and Vmz, $FA^{2-}$ concentrations are 7.55, 2.92, and 2.56 mg/kg $H_2O$ (see Table 2), and they allow $Fe^{3+}$ to be present in organo-mineral species in amounts of 84.3, 32.6, and 28.6 μg/kg $H_2O$, respectively. In this case, the total iron contents, taking into account deposition (see Tables 2 and 4), in brackish and saline waters L1, L2, and Vmz are 91, 142, and 5314 μg/kg $H_2O$.

In general, the distribution of aqueous species of Fe in the natural waters of the study area depends on the redox conditions (Eh < 0) in the aquifer and their FA content. In anoxic groundwater of the Mezen Formation, Vmz, at a depth of more than 200 m, Fe(II) predominates. The increased content of inorganic ligands in underground saline waters is the cause of a significant proportion of the complex state of Fe in the inorganic complexes $FeSO_4^0$ and $FeCl^+$ (Figure 4). The content of $FeCO_3^0$ and $FeOH^+$ is relatively low, 0.06 and 0.04%, respectively.

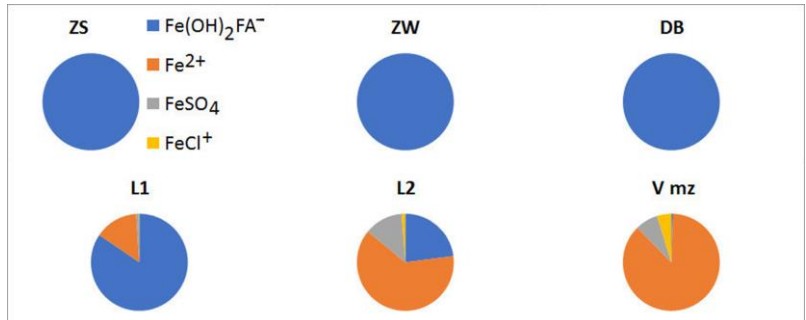

**Figure 4.** Main migratory species of Fe in the natural waters of the study area (%).

Groundwater, pumped from the "lens" in the Vendian Padun Formation, which is localized near the Lomonosovskaya pipe (L2), is also dominated by $Fe^{2+}$ (62.3%), but there is already an organo-mineral species (22.71%), $FeSO_4^0$ (12.66%), $FeCl^+$ (1.23%), $FeCO_3^0$ (1.05%), and $FeOH^+$ (0.05%). Groundwater, pumped from the "lens" in the Vendian Padun

Formation, which is localized near the Karpinskaya pipe (L1), contains more organic matter and is dominated by the fulvate complex $Fe(OH)_2FA^-$ (82.85%). The content of $Fe^{2+}$ is 14.11%, $FeCO_3^0$ is 1.92%, $FeSO_4^0$ is 0.95%, $FeCl^+$ is 014%, and $FeOH^+$ is 0.03%. In oxidizing groundwater, pumped from dewatering boreholes (Eh 65 mV) in the Vendian Padun Formation (DBs) and in the surface waters of the Zolotitsa River (ZS and ZW), all dissolved iron binds to the fulvate complex. The redox potential values in the Zolotitsa River are expectedly higher than in the dewatering boreholes, and are 120 and 280 mV in winter and summer, respectively.

**Table 4.** Simulation results of sedimentation from the surface waters of the Zolotitsa River and groundwater in the initial natural conditions.

| | ZS | ZW | DBs | L1 | L2 | Vmz |
|---|---|---|---|---|---|---|
| **Minerals, mol/kg H$_2$O** | | | | | | |
| Ankerite CaFe(CO$_3$)$_2$ | 0 | 0 | 0 | $2.22 \times 10^{-5}$ | $3.09 \times 10^{-5}$ | 0 |
| Chalcocite Cu$_2$S | 0 | 0 | 0 | $9.44 \times 10^{-9}$ | $1.45 \times 10^{-8}$ | $2.27 \times 10^{-9}$ |
| Chromite FeCr$_2$O$_4$ | 0 | 0 | 0 | $9.52 \times 10^{-9}$ | $1.48 \times 10^{-8}$ | $1.78 \times 10^{-7}$ |
| Dolomite CaMg(CO$_3$)$_2$ | 0 | 0 | 0.000196 | $1.75 \times 10^{-4}$ | 0 | 0 |
| Gibbsite Al(OH)$_3$ | $2.11 \times 10^{-6}$ | $2.11 \times 10^{-6}$ | $2.42 \times 10^{-7}$ | $9.31 \times 10^{-7}$ | $1.26 \times 10^{-6}$ | $6.93 \times 10^{-6}$ |
| Goethite FeO(OH) | $4.29 \times 10^{-7}$ | $2.85 \times 10^{-6}$ | $2.25 \times 10^{-8}$ | 0 | 0 | 0 |
| Pyrolusite MnO$_2$ | $2.44 \times 10^{-7}$ | $2.44 \times 10^{-7}$ | $2.28 \times 10^{-7}$ | 0 | 0 | 0 |
| Siderite FeCO$_3$ | 0 | 0 | 0 | 0 | 0 | $2.22 \times 10^{-5}$ |
| Sphalerite ZnS | 0 | 0 | 0 | $1.75 \times 10^{-7}$ | $5.86 \times 10^{-8}$ | $8.26 \times 10^{-8}$ |
| UO$_2$(cr) | **0** | **0** | **0** | $6.16 \times 10^{-9}$ | $6.34 \times 10^{-8}$ | $1.97 \times 10^{-10}$ |
| 2.22Galena PbS | 0 | 0 | 0 | $5.20 \times 10^{-9}$ | 0 | 0 |
| **(Ca, Sr, Zn, Pb, Mn)CO$_3$, mol/kg H$_2$O (solid solution)** | | | | | | |
| Ca | 0 | 0 | $6.01 \times 10^{-6}$ | $5.42 \times 10^{-6}$ | $3.21 \times 10^{-4}$ | 0 |
| Sr | 0 | 0 | $1.65 \times 10^{-6}$ | $4.13 \times 10^{-6}$ | $1.17 \times 10^{-4}$ | 0 |
| Zn | **0** | **0** | $6.67 \times 10^{-9}$ | $4.41 \times 10^{-9}$ | $1.03 \times 10^{-7}$ | **0** |
| Pb | 0 | 0 | $1.87 \times 10^{-10}$ | $1.71 \times 10^{-9}$ | $9.71 \times 10^{-9}$ | 0 |
| Mn | 0 | 0 | $8.18 \times 10^{-17}$ | $2.75 \times 10^{-7}$ | $8.29 \times 10^{-6}$ | 0 |

### 3.1.2. Humic Aqueous Species

With humic acids, such a sharp competitive complex formation does not occur as with fulvic acid. It is distributed more evenly among different elements. The proportion of $HA^-$ is highest in the $CaHA^+$ form, rising from 43.94% in Z to 84.37% in Vmz (Figure 3b). In the same direction, both the calcium concentration and the TDS value in the water increase. The proportion of $HA^-$ and in the form of $MgHA^+$ is significant, 15.02–32.27%. In addition, $ZnHA^+$ is present in river water at 0.33%, decreasing to 0.05–0.02% in DBs, L1, and L2 and completely absent in Vmz. $MnHA^+$ is seen in Vmz (0.22%) decreasing to 0.12–0.08% in L1 and L2. $CuHA^+$ is 0.01% in Z, and 0.02 in Vmz. $SrHA^+$ is also found in Z, L1, and L2 (0.01) as well as in Vmz (0.04%). The content of $HHA^0$ does not exceed 0.01%. At the same time, $HA^-$ in the dissociated form is also widely represented in the surface waters of the Zolotitsa River, its content being 40.67% (Figure 3b). In DBs, L1, L2, and Vmz, its content consistently decreases from 34.49 to 0.86%, "balancing" the increasing $CaHA^+$ values (Figure 5).

In the distribution structure of aqueous species of the chemical elements-macrocomponents in the natural waters of the study area, organo-mineral species play a less prominent role. Thus, the proportion of $CaHA^+$ in the Ca migratory species is only 0.01% in Z, while the proportion of $MgHA^+$ in the Mg migratory species is generally lower than 0.01% in all types of waters. In the aqueous species of trace elements, the participation of $HA^+$ is somewhat

higher. The share of ZnHA$^+$ in Zn migratory species is 0.16–0.19% in ZS–ZW and 0.01% in DBs and L1. For CuHA$^+$, the corresponding distribution is expressed as 0.14–0.31% in ZS-ZW, for PbHA$^+$ as 0.15–0.25% in ZS–ZW and 0.02% in L1, for MnHA$^+$ as 0.02% in ZS and ZW. It can be noted that the proportion of organo-mineral species with Zn, Cu, and Pb in river water is higher in winter than in summer.

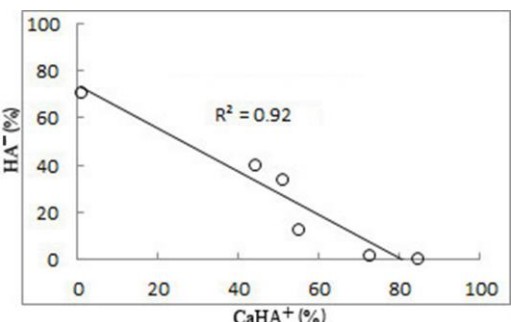

**Figure 5.** Relationship between HA$^-$ and CaHA$^+$ (%) in the natural waters of the study area. $R^2$ is the coefficient of determination.

In general, for natural surface waters of the taiga zone with sufficient alkalinity, Moiseenko et al. [42] obtained the following regularity in the predominant distribution of metals bonding to organic species: Fe > Al > Pb > Co > Ni > Zn > Cd > Cu > Mg > Ca > Cr > Mn > Sr. In our case, it looks somewhat different: Fe > Zn > Cu > Pb > Mn > Ca > Mg > Sr, probably due to the local specificity of the studied watercourse.

### 3.1.3. Inorganic Species

Elements related to the macrocomponent composition of the waters—calcium, magnesium, sodium, and potassium—migrate mainly in the ionic state. This state is typical for natural fresh waters [43]. The greatest contribution to complex formation with calcium and magnesium in fresh waters is made by hydrocarbonate, sulfate, and carbonate complexes; however, the vast majority of both elements (94–99%) are in the ionic state. In brackish and saline waters, the proportion of the latter is somewhat reduced (72–90%) due to an increase in the role of sulfate complexes (Table 3). This is due to the increase in the content of inorganic ligands in saline waters. In our case, the complexation of Ca and Mg with the sulfate ions has become important. Sodium and potassium in the groundwater of the territory are also mainly in the ionic state (more than 99% on average) [44].

Manganese in compounds exhibits oxidation states from +2 to +7, due to the specifics of the electronic configuration of the valence layer of atoms. It, like iron, belongs to the number of elements whose oxidized forms are much less soluble than the reduced ones. However, in contrast to iron, the oxidation of Mn$^{2+}$ → Mn$^{3+}$ and subsequent hydrolysis of Mn$^{3+}$ + 3OH$^-$ with the precipitation of Mn(OH) require much higher values of the redox potential (Eh > +600 mV) than those observed in the natural waters of the study area (Eh < +281 mV). In addition, unlike iron, it also has a weak ability to complex (see Section 3.1.2) and therefore migrates mainly in the form of Mn$^{2+}$ (84–98%). Due to the weak complex formation ability of manganese, even with a very high content of organic matter in natural waters, its content in ionic form is usually more than 80% [43]. According to the degree of binding of organic matter, manganese ranks below iron in different climatic zones [42]. The second dominant form of manganese occurrence in fresh waters is its carbonate complex MnHCO$_3$$^+$ (0.7–2.2%), while MnSO$_4$ (4–10%) dominates in brackish and saline waters. At the same time, according to calculations, Mn is deposited from surface waters in the amount of 13.4 μg/kg H$_2$O as part of pyrolusite, and from 12 to 455 μg/kg H$_2$O co-precipitates with calcite in groundwater DBs, L1, and L2.

Aluminum under natural pH conditions is characterized by the main forms of migration AlO$_2$$^-$ (63.76–97.7%) and AlOOH (2.29–32.13%). It is deposited in the composition of gibbsite from surface waters in the amount of 63.3 μg/kg H$_2$O. From groundwater, the

deposition rate increases from 7.26 to 208 µg/kg $H_2O$, with TDS increasing from 0.45 to 21.7 g/kg $H_2O$.

The calculations performed show a high degree of copper complexation, the dominant form of which is $CuCO_3^0$ (74.63–91.31%) in fresh waters. In relatively small amounts, copper is present in the complex states of $CuO^0$ (7.34–12.86%) and $CuOH^+$ (0.55–2.82%), as well as in the ionic state $Cu^{2+}$ (0.77–10.17%). In brackish and saline waters, the main species of copper are $CuCl^{2-}$ (24.38–80.86%) and $CuCl_3^{2-}$ (17.6–75.58%).

Zinc species are quite various. The ionic form $Zn^{2+}$ dominates (44.46–83.25%). The carbonate complex $ZnCO_3^0$ (0.06–37.52%), the hydroxo complex $ZnOH^+$ (0.65–19.02%), and the sulfate complex $ZnSO_4^0$ (0.41–23.06%) dominate. Small amounts of zinc (0.44–11.7 µg/kg $H_2O$) precipitate from groundwater as sphalerite and $CaCO_3$.

In surface and groundwater, strontium is a geochemical analog of calcium and, like calcium, is a weak complexing agent. The main species of strontium are $Sr^{2+}$ (66.76–98.44%), $SrSO_4^0$ (0.9–31.68%), and $SrHCO_3^+$ (0.04–2.35%). It co-precipitates with calcite from groundwater in significant amounts, from 0.14 to 10.3 mg/kg $H_2O$.

The behavior of uranium in water depends on the redox and acid-base conditions. In surface waters, it is in the oxidized form 6+ and migrates as part of the uranyl–carbonate complexes $UO_2(CO_3)_2^{2-}$ (55.66–57.18%) and $UO_2(CO_3)_3^{4-}$ (26.98–12.16%), as well as in the form $UO_3^0$ (14.32–24.48%), the ratio of which is determined by the acid-base conditions. In the fresh groundwater of the DBs, it is also under oxidizing conditions, but the increased alkalinity of water determines a slightly different ratio of carbonate complexes: $UO_2(CO_3)_3^{4-}$ (88.19%) and $UO_2(CO_3)^0$ (11.52%). Finally, in the reducing environment of groundwater L1, L2, and Vmz, it almost completely precipitates, and the uranium remaining in the solution migrates in the form of $UO_2OH^+$.

Nickel and cadmium migrate predominantly in the ionic state. Molybdenum predominates in the form of $MoO_4^{2-}$; for chromium, the $CrO_4^{2-}$ form is characteristic in fresh waters; in salty waters, it is in the form of $CrO^+$. Arsenic migrates in fresh waters as part of $HAsO_4^{2-}$, in saline waters—$HAsO_2^0$; lead, respectively, in the form of $PbOH^+$ and $PbCO_3^0$.

*3.2. Estimated Aqueous Organic and Inorganic Species in the Zolotitsa River When Draining Groundwater Is Discharged into It from a System of Drainage Wells*

3.2.1. Fulvic and Humic Species

The simulation results of changes in the composition of dominant migration species of chemical elements in Zolotitsa River with discharged drainage groundwater from a system of dewatering boreholes are shown in Table 5.

**Table 5.** Simulation results of changes in the composition of dominant migration forms of chemical elements in Zolotitsa River with discharged drainage groundwater from a system of dewatering boreholes.

| Aqueous Species | Scenarios of Mixing of River Waters with Drainage Groundwater | | | | | | | | | | | |
|---|---|---|---|---|---|---|---|---|---|---|---|---|
| | 1S | 1W | 2S | 2W | 3S | 3W | 4S | 4W | 5S | 5W | 6S | 6W |
| | % of Total Content | | | | | | | | | | | |
| | FA | | | | | | | | | | | |
| $Fe(OH)_2$ $FA^-$ | 100 | 100 | 100 | 100 | 100 | 100 | 100 | 100 | 100 | 100 | 100 | 100 |
| | HA | | | | | | | | | | | |
| $HA^-$ | 33.03 | 30.96 | 32.82 | 30.93 | 8.27 | 8.20 | 23.58 | 22.19 | 12.42 | 12.07 | 8.6 | 8.47 |
| $CaHA^+$ | 49.14 | 50.57 | 42.43 | 44.93 | 65.67 | 68.55 | 60.92 | 62.38 | 71.5 | 73.02 | 74.75 | 76.2 |
| $MgHA^+$ | 17.74 | 18.38 | 24.61 | 24.00 | 26.01 | 23.21 | 15.46 | 15.38 | 16.06 | 14.89 | 16.64 | 15.32 |
| $SrHA^+$ | 0.01 | 0.01 | 0.01 | 0.01 | 0.01 | 0.01 | 0 | 0 | 0 | 0 | 0 | 0 |
| $ZnHA^+$ | 0.08 | 0.08 | 0.13 | 0.13 | 0.04 | 0.04 | 0.04 | 0.05 | 0.02 | 0.02 | 0.01 | 0.01 |



| Aqueous Species | Scenarios of Mixing of River Waters with Drainage Groundwater | | | | | | | | | | | |
|---|---|---|---|---|---|---|---|---|---|---|---|---|
| | **1S** | **1W** | **2S** | **2W** | **3S** | **3W** | **4S** | **4W** | **5S** | **5W** | **6S** | **6W** |
| **% of Total Content** | | | | | | | | | | | | |
| **Al** | | | | | | | | | | | | |
| $AlO^+$ | 0 | 0.01 | 0.01 | 0.01 | 0.03 | 0.04 | 0.01 | 0.01 | 0.01 | 0.02 | 0.02 | 0.03 |
| $AlOOH$ | 2.22 | 2.54 | 2.26 | 2.56 | 4.7 | 5.06 | 2.64 | 2.97 | 3.64 | 3.98 | 4.35 | 4.71 |
| $AlO_2^-$ | 97.78 | 97.45 | 97.73 | 97.43 | 95.27 | 94.90 | 97.35 | 97.02 | 96.35 | 96 | 95.63 | 95.26 |
| **As** | | | | | | | | | | | | |
| $AsO_4^{3-}$ | 0.06 | 0.06 | 0.07 | 0.06 | 0.04 | 0.04 | 0.06 | 0.05 | 0.05 | 0.04 | 0.04 | 0.04 |
| $HAsO_4^{2-}$ | 97.99 | 97.90 | 98.07 | 98.01 | 96.47 | 96.52 | 97.7 | 97.65 | 97.03 | 97.04 | 96.63 | 96.65 |
| $H2AsO_4^-$ | 1.95 | 2.04 | 1.86 | 1.93 | 3.49 | 3.44 | 2.24 | 2.3 | 2.92 | 2.92 | 3.33 | 3.31 |
| **Ca** | | | | | | | | | | | | |
| $Ca^{2+}$ | 94.88 | 95.02 | 93.67 | 93.75 | 88.4 | 88.42 | 95.17 | 95.19 | 95.01 | 94.96 | 94.6 | 94.57 |
| $CaCl^+$ | 0.05 | 0.05 | 0.17 | 0.16 | 0.33 | 0.32 | 0.11 | 0.11 | 0.26 | 0.25 | 0.38 | 0.37 |
| $CaSO_4^0$ | 1.49 | 1.46 | 3.18 | 3.12 | 10.5 | 10.40 | 2.14 | 2.1 | 3.34 | 3.3 | 4.03 | 3.98 |
| $CaCO_3^0$ | 1.86 | 1.57 | 1.49 | 1.29 | 0.23 | 0.22 | 1.21 | 1.06 | 0.52 | 0.48 | 0.32 | 0.3 |
| $CaHCO_3^+$ | 1.72 | 1.90 | 1.49 | 1.67 | 0.54 | 0.64 | 1.37 | 1.54 | 0.87 | 1.01 | 0.67 | 0.78 |
| **Cd** | | | | | | | | | | | | |
| $Cd^{2+}$ | 88.3 | 87.70 | 69.08 | 67.42 | 51.92 | 49.73 | 77.63 | 76.34 | 59.45 | 57.41 | 49.66 | 47.52 |
| $CdOH^+$ | 0.49 | 0.32 | 0.33 | 0.22 | 0.09 | 0.06 | 0.34 | 0.22 | 0.16 | 0.11 | 0.1 | 0.07 |
| $CdCl^+$ | 10.53 | 11.28 | 29.45 | 31.09 | 45.3 | 47.13 | 21.27 | 22.62 | 38.67 | 40.52 | 47.28 | 49.02 |
| $CdCl_2^0$ | 0.07 | 0.08 | 0.72 | 0.84 | 2.54 | 2.91 | 0.33 | 0.38 | 1.5 | 1.73 | 2.79 | 3.2 |
| $CdCl_3^-$ | 0 | 0 | 0 | 0.01 | 0.03 | 0.05 | 0 | 0 | 0.01 | 0.02 | 0.04 | 0.05 |
| $CdHCO_3^+$ | 0.62 | 0.62 | 0.42 | 0.42 | 0.12 | 0.12 | 0.43 | 0.44 | 0.21 | 0.21 | 0.13 | 0.14 |
| **Cr** | | | | | | | | | | | | |
| $CrO_4^{2-}$ | 99 | 99 | 9905 | 99.05 | 98.21 | 98.31 | 98.85 | 98.87 | 98.5 | 98.56 | 98.29 | 98.37 |
| $HCrO_4^-$ | 1 | 1 | 0.95 | 0.95 | 1.79 | 1.69 | 1.15 | 1.13 | 1.5 | 1.44 | 1.71 | 1.63 |
| **Cu** | | | | | | | | | | | | |
| $Cu^{2+}$ | 0.92 | 0.94 | 1.12 | 1.13 | 6.42 | 5.87 | 1.4 | 1.39 | 3.21 | 3.01 | 5.06 | 4.68 |
| $CuOH^+$ | 0.87 | 0.64 | 0.94 | 0.68 | 2.06 | 1.43 | 1.06 | 0.76 | 1.54 | 1.08 | 1.88 | 1.31 |
| $CuO^0$ | 13.57 | 7.76 | 13.52 | 7.74 | 12.47 | 7.22 | 13.5 | 7.74 | 13.12 | 7.56 | 12.78 | 7.38 |
| $CuCl^+$ | 0 | 0 | 0.01 | 0.01 | 0.13 | 0.13 | 0.01 | 0.01 | 0.05 | 0.05 | 0.11 | 0.11 |
| $CuSO_4^0$ | 0.02 | 0.02 | 0.06 | 0.05 | 1.16 | 1 | 0.05 | 0.04 | 0.17 | 0.15 | 0.33 | 0.29 |
| $CuCO_3^0$ | 84.6 | 90.62 | 84.33 | 90.37 | 77.73 | 84.32 | 83.96 | 90.04 | 81.88 | 88.12 | 79.81 | 86.2 |
| $CuHCO_3^+$ | 0.01 | 0.01 | 0.01 | 0.01 | 0.02 | 0.02 | 0.01 | 0.01 | 0.02 | 0.02 | 0.02 | 0.02 |
| $CuHA^+$ | 0.01 | 0 | 0.01 | 0.01 | 0.01 | 0.01 | 0.01 | 0.01 | 0.01 | 0.01 | 0.01 | 0.01 |
| $CuHA_2^0$ | 0 | 0.01 | 0 | 0 | 0 | 0 | 0 | 0 | 0 | 0 | 0 | 0 |
| **Fe** | | | | | | | | | | | | |
| $Fe(OH)_2$ $FA^-$ | 100 | 100 | 100 | 100 | 100 | 100 | 100 | 100 | 100 | 100 | 100 | 100 |
| **Mg** | | | | | | | | | | | | |
| $Mg^{2+}$ | 94.93 | 95 | 92.54 | 92.69 | 82.93 | 83.44 | 94.53 | 94.58 | 93.23 | 93.38 | 92.23 | 92.41 |
| $MgOH^+$ | 0.01 | 0.01 | 0.01 | 0.01 | 0 | 0 | 0.01 | 0.01 | 0.01 | 0 | 0 | 0 |
| $MgCl^+$ | 0.08 | 0.08 | 0.28 | 0.28 | 0.53 | 0.55 | 0.18 | 0.18 | 0.43 | 0.44 | 0.63 | 0.65 |
| $MgCl_2^0$ | 0 | 0 | 0 | 0 | 0.02 | 0.02 | 0 | 0 | 0.01 | 0.01 | 0.02 | 0.02 |
| $MgSO_4^0$ | 2.32 | 2.19 | 4.94 | 4.68 | 15.89 | 15.26 | 3.32 | 3.15 | 5.2 | 4.93 | 6.29 | 5.99 |
| $MgCO_3^0$ | 0.98 | 0.87 | 0.78 | 0.71 | 0.12 | 0.12 | 0.63 | 0.58 | 0.27 | 0.26 | 0.17 | 0.17 |
| $MgHCO_3^+$ | 1.68 | 1.85 | 1.45 | 1.62 | 0.51 | 0.61 | 1.33 | 1.5 | 0.85 | 0.98 | 0.65 | 0.76 |
| **Mn** | | | | | | | | | | | | |
| $Mn^{2+}$ | 76.48 | 86.59 | 79.28 | 87.52 | 91.11 | 91.98 | 84.54 | 90.65 | 92.47 | 94.49 | 94.14 | 95.25 |

Table 5. *Cont.*

| Aqueous Species | Scenarios of Mixing of River Waters with Drainage Groundwater | | | | | | | | | | | |
|---|---|---|---|---|---|---|---|---|---|---|---|---|
| | 1S | 1W | 2S | 2W | 3S | 3W | 4S | 4W | 5S | 5W | 6S | 6W |
| **% of Total Content** | | | | | | | | | | | | |
| $MnOH^+$ | 0.11 | 0.08 | 0.1 | 0.07 | 0.04 | 0.03 | 0.09 | 0.07 | 0.07 | 0.04 | 0.05 | 0.03 |
| $MnCl^+$ | 0.04 | 0.04 | 0.16 | 0.16 | 0.38 | 0.34 | 0.11 | 0.1 | 0.28 | 0.26 | 0.42 | 0.38 |
| $MnSO_4^0$ | 0.69 | 0.74 | 1.55 | 1.63 | 6.24 | 6.04 | 1.09 | 1.12 | 1.88 | 1.83 | 2.32 | 2.23 |
| $MnHCO_3^+$ | 1.82 | 2.06 | 1.65 | 1.85 | 0.73 | 0.79 | 1.6 | 1.76 | 1.11 | 1.2 | 0.87 | 0.94 |
| $MnO_4^-$ | 20.83 | 10.48 | 17.23 | 8.76 | 1.5 | 0.82 | 12.56 | 6.3 | 4.19 | 2.18 | 2.19 | 1.17 |
| $MnO_4^{2-}$ | 0.02 | 0.1 | 0.02 | 0.01 | 0 | 0 | 0.01 | 0 | 0 | 0 | 0 | 0 |
| $MnHA^+$ | 0.01 | 0 | 0.01 | 0 | 0 | 0 | 0 | 0 | 0 | 0 | 0 | 0 |
| **Mo** | | | | | | | | | | | | |
| $MoO_4^{2-}$ | 99.99 | 99.99 | 99.99 | 99.99 | 99.99 | 99.99 | 99.99 | 99.99 | 99.99 | 99.99 | 99.99 | 99.99 |
| $HMoO_4^-$ | 0.01 | 0.01 | 0.01 | 0.01 | 0.01 | 0.01 | 0.01 | 0.01 | 0.01 | 0.01 | 0.01 | 0.01 |
| **Ni** | | | | | | | | | | | | |
| $Ni^{2+}$ | 99.89 | 99.92 | 99.87 | 99.9 | 99.89 | 99.91 | 99.9 | 99.93 | 99.89 | 99.91 | 99.88 | 99.89 |
| $NiOH^+$ | 0.1 | 0.07 | 0.09 | 0.06 | 0.03 | 0.02 | 0.08 | 0.05 | 0.05 | 0.04 | 0.04 | 0.03 |
| $NiCl^+$ | 0.01 | 0.01 | 0.04 | 0.04 | 0.08 | 0.07 | 0.02 | 0.02 | 0.06 | 0.05 | 0.08 | 0.08 |
| **Pb** | | | | | | | | | | | | |
| $Pb^{2+}$ | 0.31 | 0.30 | 0.37 | 0.35 | 1.41 | 1.25 | 0.43 | 0.4 | 0.82 | 0.74 | 1.17 | 1.05 |
| $PbOH^+$ | 44.18 | 42.78 | 46.05 | 44.36 | 65.71 | 63.21 | 49.15 | 47.2 | 58.72 | 56.2 | 63.5 | 60.99 |
| $PbO^0$ | 0.02 | 0.01 | 0.02 | 0.01 | 0.01 | 0.01 | 0.02 | 0.01 | 0.02 | 0.01 | 0.02 | 0.01 |
| $PbCl^+$ | 0.01 | 0.01 | 0.04 | 0.03 | 0.28 | 0.25 | 0.03 | 0.03 | 0.12 | 0.11 | 0.26 | 0.23 |
| $PbCl_2^0$ | 0 | 0 | 0 | 0 | 0.01 | 0.01 | 0 | 0 | 0 | 0 | 0.01 | 0 |
| $PbSO_4^0$ | 0.01 | 0.01 | 0.04 | 0.03 | 0.49 | 0.42 | 0.03 | 0.03 | 0.09 | 0.07 | 0.15 | 0.13 |
| $PbCO_3^0$ | 55.46 | 56.88 | 53.47 | 55.20 | 32.08 | 34.84 | 50.32 | 52.2 | 40.22 | 42.86 | 34.88 | 37.58 |
| $PbHA^+$ | 0.01 | 0.01 | 0.01 | 0.00 | 0.01 | 0.01 | 0.01 | 0.01 | 0.01 | 0.01 | 0.01 | 0.01 |
| **Sr** | | | | | | | | | | | | |
| $Sr^{2+}$ | 93.88 | 94.21 | 90.19 | 90.60 | 76.53 | 77.09 | 92.84 | 93.19 | 90.69 | 90.99 | 3.62 | 89.66 |
| $SrCl^+$ | 0.05 | 0.05 | 0.17 | 0.17 | 0.3 | 0.30 | 0.12 | 0.11 | 0.27 | 0.26 | 0.02 | 0.38 |
| $SrSO_4^0$ | 3.75 | 3.59 | 7.74 | 7.44 | 22.5 | 22.07 | 5.31 | 5.06 | 8.05 | 7.77 | 89.56 | 9.23 |
| $SrCO_3^0$ | 0.6 | 0.50 | 0.46 | 0.39 | 0.06 | 0.06 | 0.38 | 0.33 | 0.16 | 0.14 | 0.9 | 0.09 |
| $SrHCO_3^+$ | 1.72 | 1.65 | 1.44 | 1.40 | 0.46 | 0.48 | 1.35 | 1.31 | 0.83 | 0.84 | 5.9 | 0.64 |
| **U** | | | | | | | | | | | | |
| $UO_2OH+$ | 0 | 0 | 0 | 0 | 0.01 | 0 | 0 | 0 | 0 | 0 | 0.01 | 0 |
| $UO_2CO_3^0$ | 0.09 | 0.07 | 0.07 | 0.05 | 0.56 | 0.39 | 0.14 | 0.11 | 0.34 | 0.24 | 0.50 | 0.35 |
| $UO_2(CO_3)_2^{2-}$ | 18.9 | 15.32 | 15.02 | 12.03 | 29.96 | 23.95 | 21.0 | 17.3 | 28.1 | 22.42 | 30.27 | 24.9 |
| $UO_2(CO_3)_3^{4-}$ | 80.38 | 84.18 | 84.26 | 87.60 | 64.65 | 73.47 | 76.75 | 81.51 | 68.70 | 75.96 | 64.92 | 73.17 |
| $UO_3^0$ | 0.69 | 0.36 | 0.54 | 0.27 | 4.40 | 2.01 | 1.11 | 0.56 | 2.62 | 1.24 | 3.89 | 1.82 |
| $HUO_4^-$ | 0.15 | 0.07 | 0.11 | 0.05 | 0.42 | 0.18 | 0.20 | 0.09 | 0.33 | 0.14 | 0.41 | 0.17 |
| **Zn** | | | | | | | | | | | | |
| $Zn^{2+}$ | 42.01 | 48.14 | 45.5 | 51.33 | 66.6 | 70.64 | 49.88 | 55.76 | 63.37 | 68.71 | 69.36 | 74.09 |
| $ZnOH^+$ | 25.76 | 16.72 | 24.49 | 15.84 | 13.76 | 8.83 | 24.26 | 15.71 | 19.67 | 12.62 | 16.59 | 10.64 |
| $ZnO^0$ | 0.23 | 0.11 | 0.2 | 0.10 | 0.05 | 0.02 | 0.17 | 0.09 | 0.09 | 0.05 | 0.06 | 0.03 |
| $HZnO_2^-$ | 0.01 | 0 | 0.01 | 0 | 0 | 0 | 0 | 0 | 0 | 0 | 0 | 0 |
| $ZnCl^+$ | 0.03 | 0.03 | 0.12 | 0.10 | 0.35 | 0.28 | 0.08 | 0.07 | 0.25 | 0.2 | 0.4 | 0.32 |
| $ZnCl_2^0$ | 0 | 0 | 0 | 0 | 0.01 | 0.01 | 0 | 0 | 0 | 0 | 0.01 | 0.01 |
| $ZnSO_4^0$ | 1.06 | 1.13 | 2.48 | 2.60 | 12.67 | 12.66 | 1.79 | 1.88 | 3.58 | 3.62 | 4.73 | 4.76 |
| $ZnCO_3^0$ | 30.46 | 33.37 | 26.78 | 29.58 | 6.33 | 7.30 | 23.42 | 26.02 | 12.72 | 14.43 | 8.58 | 9.84 |
| $ZnHCO_3^+$ | 0.41 | 0.48 | 0.39 | 0.45 | 0.22 | 0.25 | 0.38 | 0.45 | 0.31 | 0.36 | 0.26 | 0.3 |
| $ZnHA^+$ | 0.03 | 0.02 | 0.03 | 0.00 | 0.01 | 0.01 | 0.02 | 0.02 | 0.01 | 0.01 | 0.01 | 0.01 |

In mixed solutions, fulvic acids, as in natural waters (Figure 3a), are completely bound to Fe in the $Fe(OH)_2FA^-$ in all six mixing scenarios. The formation of organo-mineral complexes of Fe is a classic scenario for the accumulation of Fe in natural waters.

Humic acids, as in natural waters, are distributed more evenly over different elements. The proportion of $HA^-$ is maximal in the form of $CaHA^+$, rising from 42.43 in 2S to 76.2% in 6W (Figure 6a) relative to the total HA content, which is significantly higher than in natural river waters (43.94%) (Figure 3b). The difference between summer and winter periods is insignificant (2–6%), and in winter the relative percentage of $CaHA^+$ is higher than in summer. This pattern is observed because we are considering the percentage of complexes relative to the total content of HA in the system. Since the total HA content in the river is lower in winter than in summer, most of it is complexed with metals.

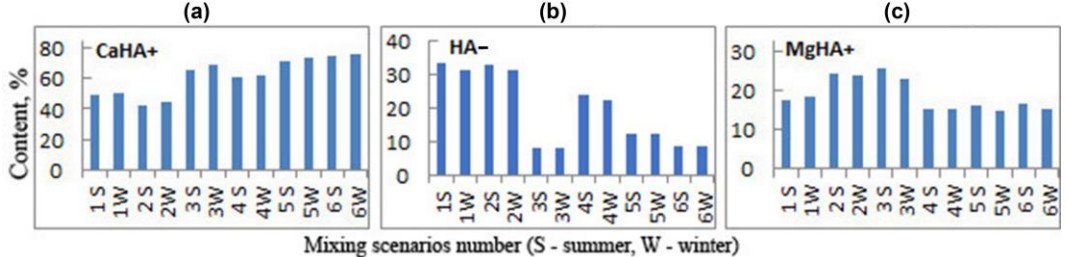

**Figure 6.** The main species of humic acids in the Zolotitsa River when draining groundwater is discharged into it from a system of drainage wells (%): (**a**) $CaHA^+$ concentrations (%) in water; (**b**) $HA^-$ concentrations (%) in water; and (**c**) $MgHA^+$ concentrations (%) in water.

The content of organo-mineral complexes in the form of $MgHA^+$ in the mixture of river waters with drainage waters (14.89–26.01%) (Figure 6c) is comparable to their content in natural river waters (15.02%). In addition, $ZnHA^+$ is present in the mixed solutions according to scenarios 1 and 2 in an amount of 0.08–0.13%, decreasing to 0.04–0.01% according to scenarios 3–6, which is significantly lower than in natural river water (0.33). $SrHA^+$ is set in the amount of 0.01% only in the mixed solutions according to scenarios 1–3. $NA^-$ in free form is contained in the amount of 30.93–33.03% in the mixed solutions according to scenarios 1 and 2, and decreases to 8.2–23.58% according to scenarios 3–6 (Figure 6b), which is significantly lower than in the natural waters of the river (40.67). This is due to an increase in the concentrations of Ca, Mg, Zn, and Sr in the mixed solutions compared to their concentrations in river waters. In other words, in the mixed solutions, metals, which mainly come from discharged groundwater, are complexed with organic matter, which is contained in river waters. The formed organo-mineral complexes are very stable, and the metals complexed in them can migrate in this form over long distances.

The percentages of $HA^-$ and $MgHA^+$ in the mixed solutions are generally lower in winter than in summer; the contents of $ZnHA^+$ and $SrHA^+$ are basically the same in summer and winter.

In the distribution structure of aqueous species of the chemical elements-macrocomponents in mixed solutions, organo-mineral complexes play a less prominent role. Thus, the proportion of $CaHA^+$ in Ca aqueous species and the proportion of $MgHA^+$ in Mg migratory species are below 0.01% in all types of waters. In the aqueous species of trace elements, $HA^+$ participation is not much higher. The proportion of $ZnHA^+$ in Zn aqueous species is 0.01–0.03%, while in natural river waters, it is in the range of 0.16–0.19%. For $CuHA^+$ and $PbHA^+$, the corresponding distribution is expressed as 0.01%, which is significantly lower than the distribution in natural river waters (0.14–0.31% and 0.15–0.25%, respectively). This is a consequence of the fact that the concentration of trace elements in mixed solutions is much higher than in river waters, while the content of organic matter, on the contrary, is lower. The proportion of $MnHA^+$ (0.01%) was established only in mixed solutions according to scenarios 1S and 2S and is close to the corresponding value in natural river waters (0.02%). At the same time, in mixed solutions, in general, the pattern noted in

Section 3.1.2 for natural river waters in the predominant distribution of metals binding to organic complexes is preserved: Fe > Zn > Cu > Pb > Mn > Ca > Mg > Sr.

### 3.2.2. Inorganic Species

For elements related to the macrocomponent composition of waters, there is a decrease in the proportion of the ionic state of calcium and magnesium in mixed solutions (82.93–95.19%) compared to natural river waters (98.49–98.93%) mainly due to an increase in the role of sulfate complexes (1.46–15.89%) pulled up to the drainage system mainly from lens L2.

The proportion of manganese in the ionic state is also lower in the mixed solutions (76.48–95.25%) compared to the natural river waters (96.83–98.49%), mainly due to an increase in the proportion of $MnO_4^-$ to 20.83% in summer and 10.48% in winter according to scenario 1, and to 17.23% in summer and 8.76% in winter according to scenario 2 (Table 5).

The proportion of the ionic state of strontium is lower in the mixed solutions (76.53–94.21%) compared to natural river waters (98.19–98.44%), mainly due to an increase in the proportion of $SrSO_4^0$ (3.59–22.66%) and $SrHCO3^+$ (0.46–5.9%) as compared with the contents of these complexes in the natural river waters (0.9–0.97% and 0.53–0.74%, respectively).

The content of aluminum in the form of $AlO_2^-$ (94.9–97.78%) in the mixed solutions is slightly higher than in the natural river waters (93.04–94.48%) due to the reduction in the proportion of $AlOOH^0$ from 5.49–8.22% to 2.22–5.06%.

The content of copper in the complex state $CuCO_3^0$ (77.73–90.62%) in the mixed solutions is also higher than in the natural river waters (74.63–80.19%) due to the reduction in the proportion of $Cu^{2+}$ from 4.77–10.17% to 0.92–5.06% and $CuO^0$ from 11.97–12.86% to 7.22–13.57%.

The ionic form of $Zn^{2+}$ accounts for 42.01–74.09% of the mixed solutions, which is significantly lower than in natural river waters (70.7–80.12%). The most significant decreases are associated with an increase in the percentage contents of $ZnCO_3^0$ (6.33–30.46%), $ZnOH^+$ (8.83–25.76%), and $ZnSO_4^0$ (1.06–12.67%) compared with the contents of these complexes in the natural river waters (4.61–9.33%, 14.32–19.02, and 0.41–0.49%, respectively).

The ratio of uranium carbonate complexes in the mixed solutions has changed due to the involvement of groundwater. The content of $UO_2(CO_3)_3^{4-}$ increased to 64.92–87.6%, and the content of $UO_2(CO_3)_2^{2-}$ decreased to 12.03–30.27% compared with the natural river waters, in which the content of $UO_2(CO_3)_3^{4-}$ is 12.16–26.98%, and the content of $UO_2(CO_3)_2^{2-}$ is 55.66–57.18%.

Nickel migrates predominantly in the ionic form in the mixed solutions. The percentage of $Cd^{2+}$ decreased due to the attraction of $CdCl^+$ from groundwater. Molybdenum still predominates in the form of $MoO_4^{2-}$; the same shape is characteristic of chromium. Arsenic still migrates mainly as $HAsO_4^{2-}$; lead, respectively, is characterized by the $PbOH^+$ and $PbCO_3$ complexes.

### 3.3. Influence of DOC on the Intensity of Precipitation of Chemical Elements from Mixed Solutions

According to the simulation results, the concentration of Fe in the solutions is higher in summer than in winter: 82–91 µg/kg $H_2O$ in winter and 127–136 µg/kg $H_2O$ in summer (Figure 7a, Table 6). This is explained by the fact that Fe in the solutions is in the form of $Fe(OH)_2FA^-$, that is, 1 mmol/kg $H_2O$ of $Fe^{3+}$ accounts for 1 mmol/kg $H_2O$ of $FA^-$, and $FA^-$ in the solutions is higher in summer and lower in winter (Table 6). Due to the strong affinity of Fe with organic matter, the entire amount of fulvic acid specified in the balance of the model is consumed for the formation of Fe organo-mineral complexes that retain Fe in the solutions. The free iron remaining in the mixed solutions precipitates in the form of goethite FeO(OH) (Table 7). Naturally, more Fe is deposited in winter than in summer (Figure 7b). Figure 6b also shows the amount of precipitating Fe calculated assuming no DOC in the mixed solutions. In this case, when about 5000 m³/h of drainage water is

discharged into the river, the mass of precipitating iron according to scenario 3 can reach 28.9 t/y. When DOC is taken into account, this mass will decrease to 21 t/y in summer and to 23.6 t/y in winter, that is, it will be 27 and 18% lower, respectively. According to the most favorable scenario 1, which will be realized during the longest periods of the deposit operation, the mass of deposited iron will be 8.8 t/y without taking into account DOC. When DOC is taken into account, this mass will decrease to 0.58 t/y in summer and to 3.55 t/y in winter, that is, it will be 93 and 60% lower, respectively.

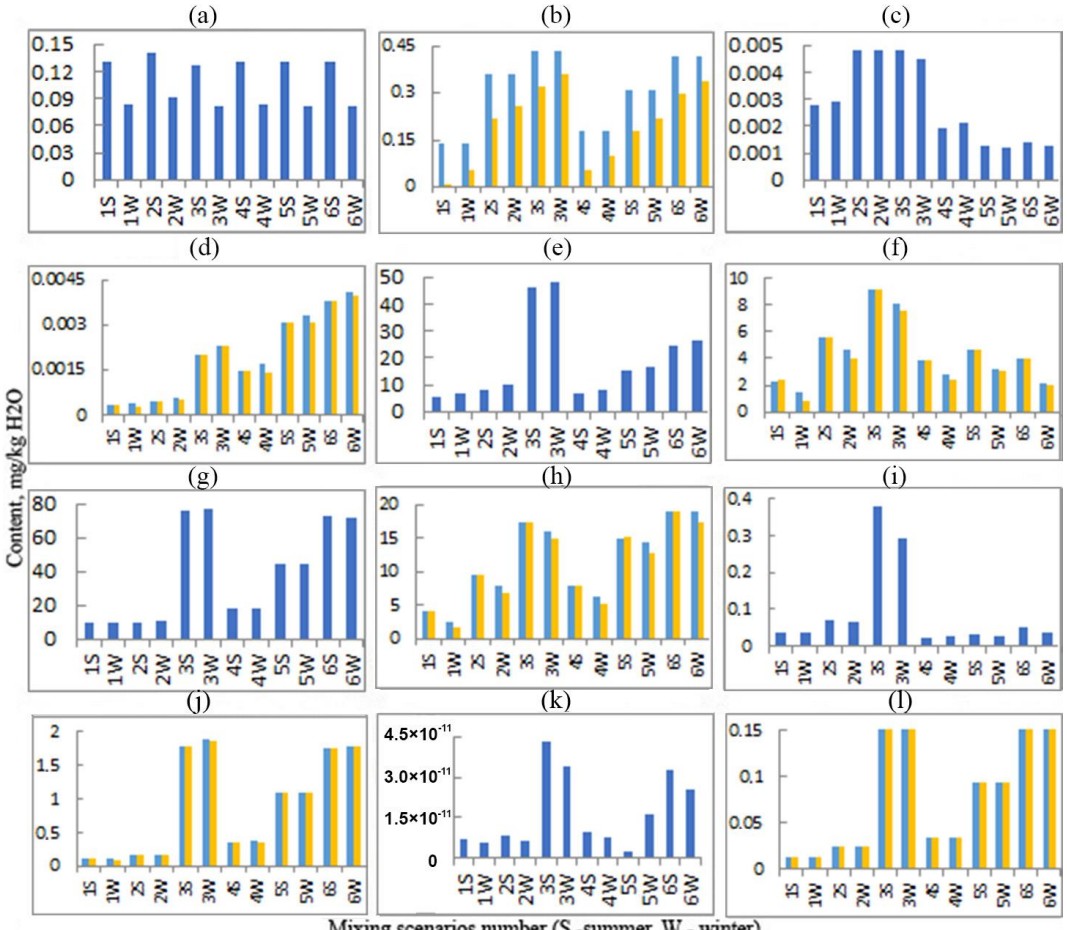

**Figure 7.** Comparison of the amounts of deposited elements with and without DOC. (**a**) Fe in the solutions; (**b**) Fe precipitated without DOC (blue) and with DOC (yellow); (**c**) Zn in the solutions; (**d**) Zn precipitated without DOC (blue) and with DOC (yellow); (**e**) Mg in the solutions; (**f**) Mg precipitated without DOC (blue) and with DOC (yellow); (**g**) Ca in the solutions; (**h**) Ca precipitated without DOC (blue) and with DOC (yellow); (**i**) Sr in the solutions; (**j**) Sr precipitated without DOC (blue) and with DOC (yellow); (**k**) Mn in the solutions; (**l**) Mn precipitated without DOC (blue) and with DOC (yellow).

**Table 6.** Simulation results of changes in the composition of water in Zolotitsa River with discharged drainage groundwater from a system of dewatering boreholes.

| | Scenarios of Mixing of River Waters with Drainage Groundwater | | | | | |
|---|---|---|---|---|---|---|
| | 1S | 1W | 2S | 2W | 3S | 3W |
| | | | mg/kg $H_2O$ | $H_2O$ | | |
| FA | 11.4 | 7.4 | 12.2 | 815 | 11.4 | 7.35 |
| HA | 0.64 | 0.412 | 0.68 | 0.45 | 0.64 | 0.41 |
| Na | 72 | 72 | 187 | 187 | 382 | 382 |

**Table 6.** *Cont.*

| | 1S | 1W | 2S | 2W | 3S | 3W |
|---|---|---|---|---|---|---|
| **Scenarios of Mixing of River Waters with Drainage Groundwater** | | | | | | |
| Mg | 5.25 | 6.76 | 8.34 | 9.97 | 46.4 | 47.9 |
| S | 7.95 | 7.95 | 22.3 | 22.3 | 135 | 135 |
| Cl | 51.4 | 51.4 | 207 | 207 | 532 | 532 |
| K | 3.09 | 3.09 | 3.53 | 3.53 | 7.98 | 7.98 |
| Ca | 9.74 | 10.2 | 9.58 | 10.3 | 76.2 | 76.6 |
| $HCO_3^-$ | 133 | 142 | 131 | 142 | 63 | 72.1 |
| $SO_4^{2-}$ | 23.9 | 23.9 | 66.9 | 66.9 | 405 | 405 |
| TDS | 310 | 318 | 625 | 635 | 1525 | 1532 |
| **µg/kg H$_2$O** | | | | | | |
| Al | 0.90 | 0.5 | 0.89 | 0.5 | 0.43 | 0.25 |
| Cr | 0.54 | 0.54 | 0.6 | 0.6 | 0.7 | 0.7 |
| Mn | $6.81 \times 10^{-9}$ | $5.38 \times 10^{-9}$ | $8.08 \times 10^{-9}$ | $6.37 \times 10^{-9}$ | $4.36 \times 10^{-9}$ | $3.4 \times 10^{-9}$ |
| Fe | 127 | 82.6 | 136 | 91 | 127 | 82.1 |
| Ni | 0.17 | 0.17 | 0.29 | 0.29 | 0.46 | 0.46 |
| Cu | 0.19 | 0.19 | 0.36 | 0.36 | 0.47 | 0.47 |
| Zn | 2.8 | 2.85 | 4.77 | 4.75 | 4.75 | 4.49 |
| As | 0.57 | 0.57 | 0.53 | 0.53 | 0.53 | 0.53 |
| Sr | 35.1 | 38.2 | 72.4 | 67.8 | 380 | 293 |
| Mo | 1.92 | 1.92 | 4.49 | 4.49 | 2.05 | 2.05 |
| Cd | 0.0039 | 0.0039 | 0.0098 | 0.0098 | 0.0098 | 0.0098 |
| Pb | 0.017 | 0.017 | 0.086 | 0.076 | 0.014 | 0.011 |
| U | 4.71 | 4.71 | 3.86 | 3.86 | 6.12 | 6.12 |

| | 4S | 4W | 5S | 5W | 6S | 6W |
|---|---|---|---|---|---|---|
| **mg/kg H$_2$O** | | | | | | |
| FA | 11.4 | 7.4 | 11.4 | 7.35 | 11.4 | 7.35 |
| HA | 0.64 | 0.41 | 0.64 | 0.41 | 0.64 | 0.41 |
| Na | 107 | 107 | 207 | 207 | 301 | 301 |
| Mg | 6.78 | 8.26 | 15 | 16.6 | 24.3 | 26.2 |
| S | 13 | 13 | 27.6 | 27.6 | 41.4 | 41.4 |
| Cl | 125 | 125 | 338 | 338 | 539 | 539 |
| K | 3.41 | 3.41 | 4.34 | 4.34 | 5.2 | 5.2 |
| Ca | 17.8 | 18.5 | 44.5 | 44.9 | 72.5 | 72.1 |
| $HCO_3^-$ | 112 | 121 | 80.8 | 90.4 | 68.6 | 76.7 |
| $SO_4^{2-}$ | 39 | 39 | 82.8 | 82.8 | 124 | 124 |
| TDS | 423 | 431 | 785 | 791 | 1147 | 1152 |
| **µg/kg H$_2$O** | | | | | | |
| Al | 0.76 | 0.43 | 0.55 | 0.32 | 0.46 | 0.27 |
| Cr | 0.66 | 0.66 | 1 | 1 | 1.32 | 1.32 |
| Mn | $9.45 \times 10^{-9}$ | $7.64 \times 10^{-9}$ | $2.03 \times 10^{-8}$ | $1.62 \times 10^{-8}$ | $3.24 \times 10^{-8}$ | $2.56 \times 10^{-8}$ |
| Fe | 127 | 82.6 | 127 | 82.1 | 127 | 82.1 |
| Ni | 0.19 | 0.19 | 0.23 | 0.23 | 0.27 | 0.27 |
| Cu | 0.19 | 0.19 | 0.19 | 0.19 | 0.2 | 0.2 |
| Zn | 1.94 | 2.1 | 1.29 | 1.23 | 1.4 | 1.24 |
| As | 0.57 | 0.57 | 0.58 | 0.58 | 0.59 | 0.59 |
| Sr | 25.1 | 28.2 | 32.8 | 27.6 | 51.6 | 39.2 |
| Mo | 1.97 | 1.97 | 2.9 | 2.09 | 2.21 | 2.21 |
| Cd | 0.0041 | 0.0041 | 0.0046 | 0.0046 | 0.0052 | 0.0052 |
| Pb | 0.003 | 0.0032 | 0.00071 | 0.00057 | 0.00048 | 0.00036 |
| U | 4.67 | 4.67 | 4.55 | 4.55 | 4.43 | 4.43 |

It should be noted that the concentration of Fe in the solutions in summer exceeds the maximum permissible concentrations (MPCs) for fishery rivers (100 µg/kg H$_2$O) established in Russia due to the increased concentrations of DOC in mixed solutions. At the

same time, in winter, the Fe content is lower than the MPCs, due to the decrease in the DOC content in the water. However, under natural conditions, the water of the Zolotitsa River contains Fe in the amount of 339 µg/kg $H_2O$, that is, drainage waters contribute to a decrease in the concentration of this element, which in this case should have a positive effect on salmon spawning. The negative impact of drainage water is associated with increased concentrations of molybdenum in them, of 2.7–18.1 µg/kg $H_2O$ (Table 2). As a result, under all scenarios, Mo concentrations exceed MPCs (1 µg/kg $H_2O$), varying from 1.92 to 4.49 µg/kg $H_2O$. Elevated concentrations of Mo in the abiotic components of the study area are associated with its transport from the Baltic (Fenno-Scandinavian) shield located to the west, where it is widely present in apatite–nepheline ores [18].

**Table 7.** Simulation results of changes in the composition of precipitations in Zolotitsa River with discharged drainage groundwater from a system of dewatering boreholes.

| | \multicolumn{6}{c}{Scenarios of Mixing of River Waters with Drainage Groundwater} | | | | | |
|---|---|---|---|---|---|---|
| | **1S** | **1W** | **2S** | **2W** | **3S** | **3W** |
| \multicolumn{7}{c}{**Phase, mol/kg $H_2O$**} | | | | | | |
| Dolomite | $9.74 \times 10^{-5}$ | $3.57 \times 10^{-5}$ | $2.35 \times 10^{-4}$ | $1.67 \times 10^{-4}$ | $3.80 \times 10^{-4}$ | $3.16 \times 10^{-4}$ |
| Gibbsite | $8.49 \times 10^{-7}$ | $8.63 \times 10^{-7}$ | $9.62 \times 10^{-7}$ | $9.76 \times 10^{-7}$ | $1.03 \times 10^{-6}$ | $1.04 \times 10^{-6}$ |
| Goethite | $1.58 \times 10^{-7}$ | $9.66 \times 10^{-7}$ | $3.90 \times 10^{-6}$ | $4.71 \times 10^{-6}$ | $5.64 \times 10^{-6}$ | $6.45 \times 10^{-6}$ |
| Pyrolusite | $2.33 \times 10^{-7}$ | $2.33 \times 10^{-7}$ | $4.34 \times 10^{-7}$ | $4.34 \times 10^{-7}$ | $2.66 \times 10^{-6}$ | $2.66 \times 10^{-6}$ |
| \multicolumn{7}{c}{**(Ca, Sr, Zn, Pb, Mn)$CO_3$, mol/kg $H_2O$ (solid solution)**} | | | | | | |
| Ca | $3.19 \times 10^{-6}$ | $2.58 \times 10^{-6}$ | $2.32 \times 10^{-6}$ | $2.37 \times 10^{-6}$ | $4.86 \times 10^{-5}$ | $5.72 \times 10^{-5}$ |
| Sr | $1.12 \times 10^{-6}$ | $1.09 \times 10^{-6}$ | $1.65 \times 10^{-6}$ | $1.71 \times 10^{-6}$ | $2.03 \times 10^{-5}$ | $2.13 \times 10^{-5}$ |
| Zn | $5.46 \times 10^{-9}$ | $4.71 \times 10^{-9}$ | $7.54 \times 10^{-9}$ | $7.75 \times 10^{-9}$ | $3.07 \times 10^{-8}$ | $3.47 \times 10^{-8}$ |
| Pb | $2.33 \times 10^{-10}$ | $2.31 \times 10^{-10}$ | $1.05 \times 10^{-9}$ | $1.09 \times 10^{-9}$ | $1.83 \times 10^{-9}$ | $1.85 \times 10^{-9}$ |
| Mn | $5.65 \times 10^{-17}$ | $3.99 \times 10^{-17}$ | $5.21 \times 10^{-17}$ | $4.44 \times 10^{-17}$ | $9.01 \times 10^{-16}$ | $8.54 \times 10^{-16}$ |
| | **4S** | **4W** | **5S** | **5W** | **6S** | **6W** |
| \multicolumn{7}{c}{**Phase, mol/kg $H_2O$**} | | | | | | |
| Dolomite | $1.63 \times 10^{-4}$ | $1.02 \times 10^{-4}$ | $1.96 \times 10^{-4}$ | $1.28 \times 10^{-4}$ | $1.64 \times 10^{-4}$ | $8.29 \times 10^{-5}$ |
| Gibbsite | $8.98 \times 10^{-7}$ | $9.09 \times 10^{-7}$ | $1.03 \times 10^{-6}$ | $1.04 \times 10^{-6}$ | $1.16 \times 10^{-6}$ | $1.16 \times 10^{-6}$ |
| Goethite | $9.30 \times 10^{-7}$ | $1.74 \times 10^{-6}$ | $3.16 \times 10^{-6}$ | $3.97 \times 10^{-6}$ | $5.26 \times 10^{-6}$ | $6.07 \times 10^{-6}$ |
| Pyrolusite | $6.10 \times 10^{-7}$ | $6.10 \times 10^{-7}$ | $1.70 \times 10^{-6}$ | $1.70 \times 10^{-6}$ | $2.73 \times 10^{-6}$ | $2.73 \times 10^{-6}$ |
| \multicolumn{7}{c}{**(Ca, Sr, Zn, Pb, Mn)$CO_3$, mol/kg $H_2O$ (solid solution)**} | | | | | | |
| Ca | $3.06 \times 10^{-5}$ | $2.42 \times 10^{-5}$ | $1.81 \times 10^{-4}$ | $1.88 \times 10^{-4}$ | $3.09 \times 10^{-4}$ | $3.51 \times 10^{-4}$ |
| Sr | $2.34 \times 10^{-8}$ | $4.10 \times 10^{-6}$ | $1.24 \times 10^{-5}$ | $1.25 \times 10^{-5}$ | $2.01 \times 10^{-5}$ | $2.02 \times 10^{-5}$ |
| Zn | $4.13 \times 10^{-6}$ | $2.09 \times 10^{-8}$ | $4.70 \times 10^{-8}$ | $4.80 \times 10^{-8}$ | $5.83 \times 10^{-8}$ | $6.07 \times 10^{-8}$ |
| Pb | $3.01 \times 10^{-10}$ | $3.00 \times 10^{-10}$ | $3.14 \times 10^{-10}$ | $3.14 \times 10^{-10}$ | $3.16 \times 10^{-10}$ | $3.17 \times 10^{-10}$ |
| Mn | $4.54 \times 10^{-16}$ | $3.08 \times 10^{-16}$ | $2.53 \times 10^{-15}$ | $2.18 \times 10^{-15}$ | $4.30 \times 10^{-15}$ | $4.04 \times 10^{-15}$ |

The removal of $OH^-$ from the solution proceeds similarly to the precipitation of Fe (Table 7). The concentration of $OH^-$, which is in the solutions as part of the $Fe(OH)_2FA^-$ complex compound, is also proportional to the concentrations of $FA^-$.

The amount of precipitating zinc is two orders of magnitude lower than the amount of precipitating iron and one order of magnitude lower than the amount of dissolved zinc (Figure 7c,d). The mass of Zn precipitating in the composition of the calcite (Table 7) according to scenario 6 can reach 0.19 t/y and practically does not depend on the season, since the share of $ZnHA^+$ in the total balance of zinc compounds is negligible.

In contrast to Fe, the concentrations of Ca, Mg, and C in the solutions with DOC are somewhat higher in winter than in summer: 10.2–76.5, 6.76–47.9, and 14.2–28 mg/kg $H_2O$ and 9.55–76.2, 5.25–46.4, and 12.4–26.1 mg/kg $H_2O$, respectively (Figure 7e,g, Table 6). At the same time, the summer concentrations of these elements in the solutions with DOC are identical to the concentrations in the solutions without DOC [25]. Accordingly, their

precipitation is somewhat worse in winter than in summer. Ca precipitates in summer from 4 to 18.9 mg/kg $H_2O$, and in winter from 1.53 to 17.4 mg/kg $H_2O$; Mg precipitates in summer from 2.33 to 9.12 mg/kg $H_2O$, and in winter from 0.86 to 7.58 mg/kg $H_2O$; and C precipitates in summer from 2.39 to 9.96 mg/kg $H_2O$, and in winter from 0.9 to 8.52 mg/kg $H_2O$. At the same time, in summer, the precipitation of these elements is the same both in the presence of DOC and in its absence (Figure 7f,h, Table 7).

Strontium in the solutions is higher in summer, and correspondingly more precipitated in winter (Figure 7i,j). Manganese is practically absent in the solutions, and its precipitation is practically independent of the season (Figure 7k,l).

Figure 8 shows a predictive model for the transformation of the natural waters of the Zolotitsa River under the influence of drainage waters under various scenarios.

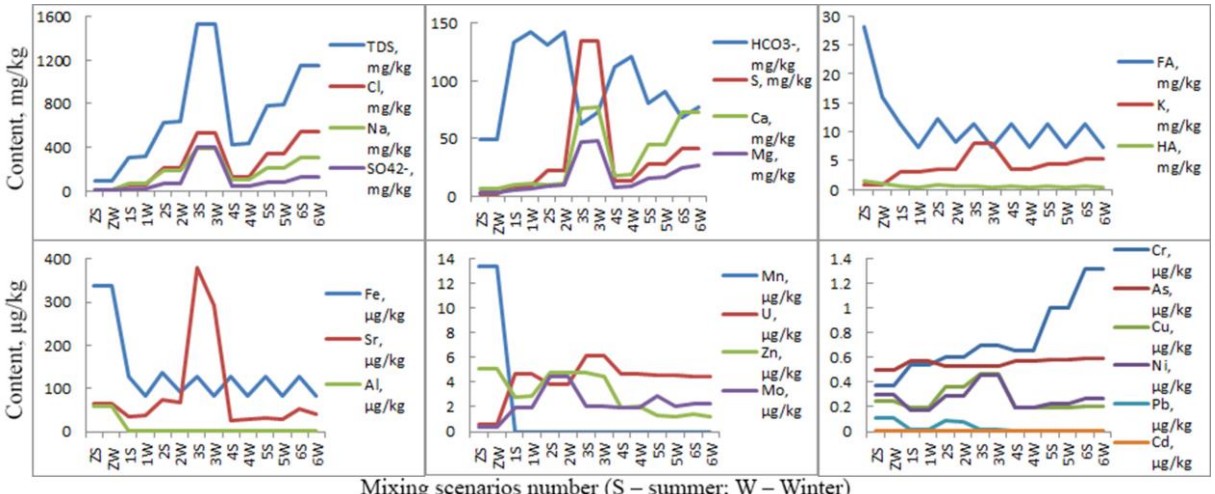

**Figure 8.** Predictive model for the transformation of the natural waters of the Zolotitsa River (ZS, ZW) under the influence of drainage waters (1S−6W).

## 4. Conclusions

It has been established that fulvic acid completely binds to Fe in the $Fe(OH)_2FA^-$ complex in all types of natural waters and under all mixing scenarios. With humic acid, such a sharp competitive complex formation does not occur. It is distributed among the various elements more evenly. The distribution of Fe species in natural waters depends on the content of $FA^{2-}$ in them and on the redox conditions in the aquifer. The main forms in brackish and saline waters are $Fe^{2+}$ and the inorganic complexes $FeSO_4$ and $FeCl^+$.

It has been determined that the mass of precipitating iron in the absence of DOC in the mixed solutions can reach 28.9 tons per year. In the presence of DOC, this weight is reduced by 18–27%. According to the most favorable scenario, the mass of deposited iron, excluding DOC, will be 8.8 tons per year. When taking into account DOC, it will be lower by 60–93%.

The amount of precipitating zinc is two orders of magnitude lower than the amount of precipitating iron and one order of magnitude lower than the amount of dissolved zinc. The mass of precipitated Zn can reach 0.19 tons per year and practically does not depend on the season, since the share of $ZnHA^+$ in the total balance of zinc compounds is negligible.

Comparison of the data obtained in the calculations with allowance for organic matter with data without it showed that fulvic and humic acids make a significant contribution to the distribution of metals between the aqueous phase and bottom sediments. A high DOC content in water helps to retain metals in the water, so calculations in such environments must be made taking DOC into account.

This study contributes to a better understanding of the behavior of heavy metals in surface waters and sediments under anthropogenic pressures in order to improve the sustainable management of water resources in the face of anthropogenic activities.

**Author Contributions:** Conceptualization, formal analysis, and writing—original draft preparation, A.I.M.; methodology and investigation, E.S.S. and E.V.C. All authors have read and agreed to the published version of the manuscript.

**Funding:** This work was supported by the Russian Ministry of Education and Science (project no. 122011300333-1). The publication was supported by the Russian Science Foundation and the Government of the Arkhangelsk Region (project No. 23-27-10004).

**Data Availability Statement:** The data presented in this study are available on request from the corresponding author.

**Conflicts of Interest:** The authors declare no conflict of interest. The founding sponsors had no role in the design of the study; in the collection, analyses, or interpretation of data; in the writing of the manuscript, and in the decision to publish the results.

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
