# Peer review of "The Influence of DOC on the Migration Forms of Elements and Their Sedimentation from River Waters at an Exploited Diamond Deposit (NW Russia)"

_water, doi:10.3390/w15122160_

Round 1
Reviewer 1 Report
In this article, a diamond deposit is selected, developed by quarries of great depth, and a forecast is made of the impact of drainage water discharge on changes in the composition of surface water and bottom sediments during the entire period of development of the deposit. The data in this article is sufficient and reliable, and the logic is rigorous and rigorous.This study contributed to a better understanding of the behavior of heavy metals in surface waters and sediments under anthropogenic pressures in order to improve the sustainable management of water resources in the face of anthropogenic activities. The technical methods of the manuscript are correct and the results are reliable. It is recommended to accept .
Author Response
Reviewer 1
In this article, a diamond deposit is selected, developed by quarries of great depth, and a forecast is made of the impact of drainage water discharge on changes in the composition of surface water and bottom sediments during the entire period of development of the deposit. The data in this article is sufficient and reliable, and the logic is rigorous. This study contributed to a better understanding of the behavior of heavy metals in surface waters and sediments under anthropogenic pressures in order to improve the sustainable management of water resources in the face of anthropogenic activities. The technical methods of the manuscript are correct and the results are reliable. It is recommended to accept.
The authors are grateful to the reviewer for the positive assessment of their work.

Reviewer 2 Report
In this article, a diamond deposit is selected, developed by quarries of great depth, and a forecast is made of the impact of drainage water discharge on changes in the composition of surface water and bottom sediments during the entire period of development of the deposit. The authors are advised to consider the following suggestions to further improve the paper quality.
1) When describing the calculation of precipitated iron mineral content, it is recommended to explain the specific calculation method, such as the model and formula used. This can make the article more detailed and reliable.
2) It is recommended to use more standardized chemical terms when describing the existence form of elements, such as "ionic state", "complex state", "oxidized state", etc.
3) It is recommended to provide more specific conditions and mechanisms when describing the precipitation and deposition of elements, such as the range of redox potential values required for precipitation, and the calculation method of deposition rate.
4) It is recommended to provide more explanations and analysis when describing the migration and distribution of elements, such as analyzing the differences in the forms of elements in different water bodies and the competitive relationships between elements.
5) In section 3.1.3, it is recommended to provide more information on the mechanism of migration of elements such as calcium, magnesium, sodium, and potassium in the form of ion dissolution. At the same time, it is recommended to provide a more detailed description to help readers understand the migration patterns of different elements in different types of water bodies.
6) It is mentioned in the text that the element content of organic mineral complexes in the mixed solution is lower than that of natural river water. However, in a mixed solution, metals mix with organic compounds in river water to form stable organic mineral complexes, which can migrate over long distances in this form. It is recommended to provide more detailed information to explain the stability and migration of these complexes in mixed solutions.
7) It is mentioned in the text that the contribution of organic mineral complexes to trace element content is not so significant. However, it is recommended to further explore the interactions between these complexes and trace elements to help readers understand the contribution of these complexes to trace elements.
8) It is recommended to provide a more detailed description to help readers understand the interactions between different elements mentioned in the text, such as the elements such as iron and manganese mentioned in section 3.1.3.
It is recommended to modify the sentences in this article to make readers read more smoothly.
Author Response
Reviewer 2
In this article, a diamond deposit is selected, developed by quarries of great depth, and a forecast is made of the impact of drainage water discharge on changes in the composition of surface water and bottom sediments during the entire period of development of the deposit. The authors are advised to consider the following suggestions to further improve the paper quality.
1) When describing the calculation of precipitated iron mineral content, it is recommended to explain the specific calculation method, such as the model and formula used. This can make the article more detailed and reliable.
We have added short information about the algorithms used in the program HCh. Analytically determined concentrations of chemical elements were given as input data to the model. The equilibrium compositions of natural waters and mixing solutions were calculated in the proportions described above. The equilibria in the program are calculated by the Gibbs free-energy minimization algorithm. The calculation of the activity coefficients of the components of the aqueous solution is carried out according to the Debye-Huckel model.
2) It is recommended to use more standardized chemical terms when describing the existence form of elements, such as "ionic state", "complex state", "oxidized state", etc.
Thank you for the specific comments, we have made changes in accordance with them. All changes are highlighted in green color.
3) It is recommended to provide more specific conditions and mechanisms when describing the precipitation and deposition of elements, such as the range of redox potential values required for precipitation, and the calculation method of deposition rate.
We added information about redox potential values. We are grateful that you brought this to your attention. Redox potential values is the key aspect in the Fe behavior in studied waters.
We did not take into account the deposition rate, so we do not provide information about this process. This would probably be relevant when taking into account the flow rate. But now we have considered a static system, since this was the first calculation taking into account organic matter at this site. Therefore, we limited solving the problem of calculating the equilibrium compositions of natural waters from different objects and also their mixtures in various proportions.
4-5) It is recommended to provide more explanations and analysis when describing the migration and distribution of elements, such as analyzing the differences in the forms of elements in different water bodies and the competitive relationships between elements. In section 3.1.3, it is recommended to provide more information on the mechanism of migration of elements such as calcium, magnesium, sodium, and potassium in the form of ion dissolution. At the same time, it is recommended to provide a more detailed description to help readers understand the migration patterns of different elements in different types of water bodies.
Thank you for the specific comments, we have made changes in accordance with them. All changes are highlighted in green color.
6) It is mentioned in the text that the element content of organic mineral complexes in the mixed solution is lower than that of natural river water. However, in a mixed solution, metals mix with organic compounds in river water to form stable organic mineral complexes, which can migrate over long distances in this form. It is recommended to provide more detailed information to explain the stability and migration of these complexes in mixed solutions.
This is true. If we consider the distribution of organic mineral species of elements relative to the content of organic matter, then it is obvious that the content of humic forms of metals in mixing solutions is lower than in groundwater. This is due to the fact that there is much more organic matter in river waters than in underground waters and most of it is in free form. Generally, it can be considered as some potential for the accumulation of heavy metals.
7) It is mentioned in the text that the contribution of organic mineral complexes to trace element content is not so significant. However, it is recommended to further explore the interactions between these complexes and trace elements to help readers understand the contribution of these complexes to trace elements.
In fact, organo-mineral species are not dominant in the forms distribution of most metal. This does not apply to iron. The accumulation of iron and, in general, its behavior in the system under consideration completely depends on the existence of organic matter.
8) It is recommended to provide a more detailed description to help readers understand the interactions between different elements mentioned in the text, such as the elements such as iron and manganese mentioned in section 3.1.3.
Thank you for the specific comments, we have made changes in accordance with them. All changes are highlighted in green color.

Reviewer 3 Report
The approach used and the results obtained significantly expand the understanding of the transformation of natural waters and the migration of heavy metals in the considered area of the development of a diamond deposit.
At the same time, it is necessary to note the shortcomings that it is desirable to eliminate in the final version of the article.
1. Figure 1 needs to be edited. This remark especially applies to Figure 1b, which does not contain the iD3-C1 object indicated in the legend. In the legend, you need to give explanations for all dashed and solid lines of different colors, vertical lines and circles with letter and number inscriptions.
2. L51-54. The figure caption to Figure 1 contains elements of the hydrogeological model of the area, which are not shown in the figure, in particular, the mention of saline waters. I recommend highlighting a separate section in the text describing the geological, hydrological and hydrogeological conditions of the area. Such a section characterizing the natural conditions of the study area would be very useful when comparing the natural conditions at various diamond deposits.
3. L229-230. Figure 4. In the caption, you need to give an explanation of R2.
To improve the perception of the results of the study, it is desirable to present a predictive model for the transformation of natural and drainage waters in a graphical or tabular form.
Author Response
Reviewer 3
The approach used and the results obtained significantly expand the understanding of the transformation of natural waters and the migration of heavy metals in the considered area of the development of a diamond deposit.
At the same time, it is necessary to note the shortcomings that it is desirable to eliminate in the final version of the article.
- Figure 1 needs to be edited. This remark especially applies to Figure 1b, which does not contain the iD3-C1 object indicated in the legend. In the legend, you need to give explanations for all dashed and solid lines of different colors, vertical lines and circles with letter and number inscriptions.
Thanks for the specific comment.
I have shown in Figure 1b the location of a kimberlite pipe with index iD3-C1. In the legend, I gave explanations for all the dashed and solid lines of different colors, vertical lines and circles with letter and number inscriptions.
- L51-54. The figure caption to Figure 1 contains elements of the hydrogeological model of the area, which are not shown in the figure, in particular, the mention of saline waters. I recommend highlighting a separate section in the text describing the geological, hydrological and hydrogeological conditions of the area. Such a section characterizing the natural conditions of the study area would be very useful when comparing the natural conditions at various diamond deposits.
Thanks for the specific comment.
I have highlighted in the text a description of the geological, hydrological and hydrogeological conditions of the region as a separate section: 2.1. Natural conditions of the study area
- L229-230. Figure 4. In the caption, you need to give an explanation of R2.
In the caption to Figure 5, I gave an explanation of R2: R2 is the coefficient of determination
- To improve the perception of the results of the study, it is desirable to present a predictive model for the transformation of natural and drainage waters in a graphical or tabular form.
Thanks for the specific comment.
I presented a predictive model for the transformation of natural and drainage waters in a graphical form:
Figure 8. Predictive model for the transformation of natural and drainage waters
